# Molecular insights into Spindlin1-HBx interplay and its impact on HBV transcription from cccDNA minichromosome

Wei Liu [1,2,9], Qiyan Yao[3,4,9], Xiaonan Su[1,2], Yafang Deng[1], Mo Yang[5,7], Bo Peng[3], Fan Zhao [1,2], Chao Du[1,2], Xiulan Zhang[1], Jinsong Zhu[5,8], Daliang Wang [1] ✉, Wenhui Li [3,6] ✉ & Haitao Li [1,2] ✉

Molecular interplay between host epigenetic factors and viral proteins constitutes an intriguing mechanism for sustaining hepatitis B virus (HBV) life cycle and its chronic infection. HBV encodes a regulatory protein, HBx, which activates transcription and replication of HBV genome organized as covalently closed circular (ccc) DNA minichromosome. Here we illustrate how HBx accomplishes its task by hijacking Spindlin1, an epigenetic reader comprising three consecutive Tudor domains. Our biochemical and structural studies have revealed that the highly conserved N-terminal 2–21 segment of HBx (HBx$_{2-21}$) associates intimately with Tudor 3 of Spindlin1, enhancing histone H3 "K4me3-K9me3" readout by Tudors 2 and 1. Functionally, Spindlin1-HBx engagement promotes gene expression from the chromatinized cccDNA, accompanied by an epigenetic switch from an H3K9me3-enriched repressive state to an H3K4me3-marked active state, as well as a conformational switch of HBx that may occur in coordination with other HBx-binding factors, such as DDB1. Despite a proposed transrepression activity of HBx$_{2-21}$, our study reveals a key role of Spindlin1 in derepressing this conserved motif, thereby promoting HBV transcription from its chromatinized genome.

Despite the existence of an effective prophylactic vaccine, Hepatitis B virus (HBV) infection continues to be a major health problem of global impact with at least 250 million people chronically infected worldwide[1,2]. HBV is a partially double-stranded relaxed circular DNA (rcDNA) virus, which enters the hepatocytes through a liver-specific receptor, sodium taurocholate cotransporting polypeptide (NTCP)[3], and chronic HBV infection could lead to liver hepatitis, cirrhosis, eventually, hepatocellular carcinoma (HCC). Upon infection, the rcDNA is transported into the nucleus and converted into an episomal covalently closed circular (ccc) DNA, which serves as the transcription template for all HBV viral RNAs, including the 3.5 kb pregenome (pg) RNA and preC mRNA, 2.4 kb preS mRNA, 2.1 kb S mRNA, and 0.7 kb HBx mRNA. In the host, the episomal cccDNA is believed to be organized into a minichromosome by histone and non-histone proteins and constitutes a persistent viral reservoir liable for viral relapses[4–8]. However, the molecular mechanisms controlling cccDNA chromatinization and its gene expression are still poorly understood.

[1]State Key Laboratory of Molecular Oncology, MOE Key Laboratory of Protein Sciences, Beijing Frontier Research Center for Biological Structure, SXMU-Tsinghua Collaborative Innovation Center for Frontier Medicine, School of Medicine, Tsinghua University, Beijing 100084, China. [2]Tsinghua-Peking Center for Life Sciences, Beijing 100084, China. [3]National Institute of Biological Sciences, Beijing 102206, China. [4]Graduate School of Peking Union Medical College, Chinese Academy of Medical Sciences, Beijing 100730, China. [5]National Center for Nanoscience and Technology, Beijing 100190, China. [6]Tsinghua Institute of Multidisciplinary Biomedical Research, Tsinghua University, Beijing 100084, China. [7]Present address: Chemical Biology Laboratory, National Cancer Institute, 1050 Boyles Str., Frederick, MD 21702, USA. [8]Present address: Suzhou Puxin Life Science Technology, Ltd, Suzhou 215124, China. [9]These authors contributed equally: Wei Liu, Qiyan Yao. ✉e-mail: wangdaliang@tsinghua.edu.cn; liwenhui@nibs.ac.cn; lht@tsinghua.edu.cn

The HBV genome encodes the polymerase (P), core (C), surface (S), and X genes. The HBV regulatory protein X (HBx), a product of the X gene, has been reported to play an important role in HBV life cycle through interaction with many host proteins. The best characterized HBx binding partner is the damage-specific DNA-binding protein 1 (DDB1), an adapter protein for the Cul4A E3 ligase complex[9]. The HBx binds to DDB1 through an H-box motif, which redirects the DDB1-contaning E3 ubiquitin ligase to degrade the structural maintenance of chromosomes (SMC) 5/6 proteins localized to Nuclear Domain 10 (ND10, a.k.a. PML-NBs), thereby relieving the HBV transcription inhibition by SMC5/6 to allow productive HBV replication[10–15]. HBx interacts with Bcl-2 or Bcl-xL through the BH3-like domain to promote HBV production[16,17]. Also, HBx activates HBV transcription through the inhibition of cellular factors involved in transcriptional and chromatin regulation, such as the PP1/HDAC1 complex and PRMT1[18,19]. Additionally, HBx protein plays a key role in the establishment of an active cccDNA chromatin state by recruitment of chromatin modifying enzymes[6,20–22]. While cccDNA apparently can be transcribed in the absence of HBx to produce HBx itself upon infection[23], in the absence of HBx, the viral genome exists in a repressed chromatin state marked by hypoacetylation and histone H3K9 methylation, correlating with the recruitment of histone deacetylases HDAC1 and H3K9 methyltransferase SETDB1, as well as the recruitment of the heterochromatin protein HP1[20]. By contrast, the existence of HBx can relieve the repressive state of HBV cccDNA minichromosome through the recruitment of the p300 acetyltransferase and histone H3K9me1/2 demethylase lysine-specific demethylase-1 (LSD1) and H3K4me3 methyltransferase Set1A[21]. Recently, Spindlin1 was identified as an HBx binding partner and was proposed to serve as an intrinsic antiviral defense factor through inhibiting HBV transcription[24], however, the underlying mechanism and structural basis for the interaction have not been determined.

Spindlin1 is a transcriptional coactivator that contains three tandem Tudor domains[25], of which the second Tudor recognizes H3K4me3, thereby stimulating target gene expression, such as rRNA genes[26,27]. Our previous studies revealed that Spindlin1 also serves as a potent reader of H3 "K4me3-R8me2a" and "K4me3-K9me3" histone methylation patterns owing to additional H3R8me2a or H3K9me3 mark readout by Tudor 1, thus promoting Wnt/TCF4 signaling or gene expression at H3K9me3-enriched heterochromatic regions, respectively[28,29]. Recently, we and others reported that a Spindlin1 binding partner, SPINDOC (a.k.a. C11orf84), modulates the transcriptional coactivator activity of Spindlin1 through direct interaction with its third Tudor[30,31]. As a proto-oncogene, Spindlin1 is overexpressed and promotes oncogenic transcriptional programs in multiple types of malignant tumors, including ovarian cancer, colorectal cancer, breast cancer, liposarcoma, and HCC[32,33]. Given the coactivator role of Spindlin1, it remains an intriguing question to explore how Spindlin1 inhibits or regulates HBV transcription.

Here, we perform biochemical and crystal structural studies to explore the molecular basis underlying Spindlin1-HBx interplay. We demonstrate that Spindlin1 directly binds to a conserved N-terminal motif of HBx through its third Tudor, which is compatible with histone methylation readout by Tudors 2 and 1. Interestingly, our functional studies reveal that Spindlin1 promotes, but not inhibits, HBV transcription from the cccDNA minichromosome in a manner dependent on HBx, DDB1 activity on SMC6, and H3 "K4me3-K9me3" readout. Hence, our work provides new insights into the molecular function of Spindlin1 in sustaining HBV life cycle and its chronic infection by regulating transcription of chromatinized HBV genome.

## Results

### HBx recruits Spindlin1 through a conserved N-terminal motif
HBx is composed of 154 residues that are organized into an N-terminal negative regulatory region (residues 1 to 50) and a C-terminal transactivation region (residues 51 to 154). Characteristic motifs within HBx include a highly conserved "A" region[34], a Ser/Pro-rich region[35], an H-box motif[12], a BH3-like motif[16], and a zinc finger motif formed by conserved Cys/His residues (C61, C69, C137, and H139)[36] (Fig. 1a). To map the HBx regions responsible for Spindlin1 interaction, we first synthesized peptide array covering full HBx sequence with 10 overlapping residues. Then, we immobilized the peptides onto a 3D-carbene chip and performed lab-on-chip screening of purified Spindlin1$_{50–262}$ sample using an SPRi platform[37]. Intriguingly, peptide HBx$_{2–21}$ (the first Met residue is removed according to N-terminal Met excision rule[38]) displayed the strongest binding signal, followed by HBx$_{11–30}$ and HBx$_{61–80}$ peptides (Fig. 1b). As a positive control, we also confirmed H3K4me3 peptide binding by Spindlin1 during screening.

We next performed thermal shift assay (TSA) and isothermal titration calorimetry (ITC) experiments to validate these interactions. As anticipated, Spindlin1$_{50–262}$ displayed the most pronounced stabilization effect on HBx$_{2–21}$ peptide by +12 °C (Fig. 1c). We also measured a dissociation constant ($K_d$) of 1.1 µM (Fig. 1d) between Spindlin1$_{50–262}$ and HBx$_{2–21}$, which is about 44-fold and 55-fold stronger than that to HBx$_{11–30}$ and HBx$_{61–80}$ peptides, respectively (Supplementary Fig. 1a). Then, the ability of Spindlin1 to interact with HBx was confirmed in HEK 293 T cells by immunoprecipitation assay (Fig. 1e). The binding affinity of Spindlin1 to mutant HBx$_{2-21}$, which contains alanine mutations spanning amino acids 2 to 21 residues, was significantly decreased in comparison with wild-type HBx (Fig. 1f), confirming that HBx$_{2-21}$ residues are required for Spindlin1 interaction. In addition, mutant HBx$_{11–30}$ or HBx$_{61–80}$ also showed a weaker interaction with Spindlin1 (Supplementary Fig. 1b-d). All three mutant HBx proteins were expressed at similar levels as the wild-type HBx (Input, Fig. 1f and Supplementary Fig. 1c), implying that the failure of HBx to interact with Spindlin1 was not because of instability of the mutant proteins. Altogether, these results suggested that Spindlin1 is a cellular partner for HBx and directly interacts with 20 residues at the N-terminus of HBx.

### Structure of Spindlin1 bound to HBx$_{2-21}$ peptide
Spindlin1 contains 262 amino acids and is composed of an N-terminal flexible tail and triple Tudor repeats (Fig. 2a). To gain molecular insight into Spindlin1-HBx interaction, we solved the crystal structure of Spindlin1$_{50–262}$ bound to HBx$_{2–21}$ peptide at 1.8 Å (Fig. 2b and Table 1). Spindlin1$_{50–262}$ are organized into an integrated triangular architecture. Interestingly, HBx$_{2–21}$ adopts a "β-hairpin" structure and interacts with Tudor 3 of Spindlin1 to complete its β-barrel fold (Fig. 2b, c). Contact surface analysis revealed that HBx$_{2–21}$ is intimately associated with the hydrophobic groove of Tudor 3 with a high degree of shape complementarity (Sc = 0.7) (Fig. 2d), which is comparable to that of antibody/antigen interfaces (Sc ranges from 0.64 to 0.68)[39].

Recently, SPINDOC$_{256–281}$ was also shown to interact with Spindlin1 through Tudor 3[31]. However, structural alignment revealed that HBx$_{2–21}$ engages with Tudor 3 of Spindlin1 in an opposite N to C orientation as compared to that of SPINDOC$_{256–281}$ (Fig. 2e). Despite this, both HBx$_{2–21}$ and SPINDOC$_{256–281}$ complete the β-barrel fold of Tudor 3 with similar hydrophobic core and β-sheet formation (Supplementary Fig. 2a). This highlights both the conservation and diversity of Spindlin1's Tudor 3-mediated partner engagement. Upon complex formation, HBx$_{2–21}$ triggered minimal conformational change of Tudor 3 (Supplementary Fig. 2b), while SPINDOC$_{256–281}$ induced more pronounced structural rearrangement around β strands 1 and 2 of Tudor 3 (Supplementary Fig. 2c). One reason for the observed difference is the unique extended N-terminal motif of SPINDOC$_{256–281}$, starting from F256, which contributes to binding (Supplementary Fig. 2c).

### Details of Spindlin1-HBx$_{2–21}$ interaction and mutagenesis studies
In the co-crystal structure, the HBx segment is deeply anchored into Spindlin1 via extensive hydrophobic interactions, involving residues

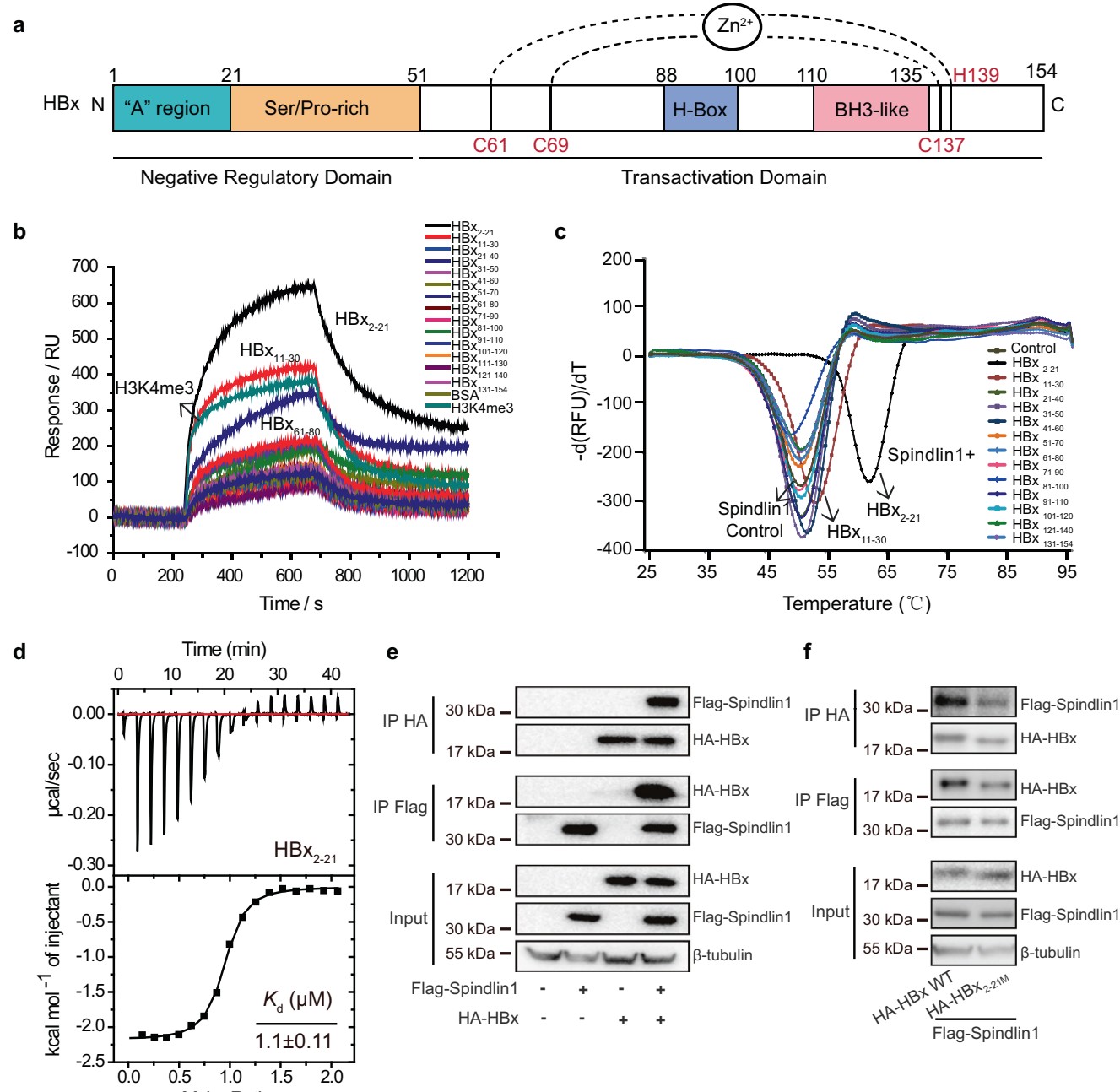

**Fig. 1 | HBx recruits Spindlin1 through a conserved N-terminal motif. a** HBx protein functional domains organization. The "A" region, Ser/Pro-rich region, H-box motif, and BH3-like motif are colored cyan, orange, blue and pink, respectively. HBx coordinates zinc via the conserved cysteine residues 61, 69,137 and histidine residues 139. **b** Surface plasmon resonance imaging profiling of Spindlin1$_{50-262}$ protein binding to different HBx peptides and H3K4me3 peptide. **c** Thermo-fluor shift melting curves of Spindlin1$_{50-262}$ with different HBx peptides. **d** ITC fitting curves of HBx$_{2-21}$ peptide titrated to human Spindlin1$_{50-262}$ protein. **e** Co-immunoprecipitation (Co-IP) of Flag-Spindlin1 with HA-HBx using anti-Flag M2 agarose beads and anti-HA antibodies in HEK 293 T cells. Whole-cell lysates and immunoprecipitates were subjected to WB with the indicated antibodies. **f** Determination of the HBx$_{2-21}$ region interacting with Spindlin1. HEK 293 T cells were co-transfected with Flag-Spindlin1 and HA-tagged WT HBx or mutant HBx containing alanine substitution spanning amino acids 2 to 21 (HA-HBx$_{2-21M}$). Cellular extracts were immunoprecipitated with anti-Flag M2 beads and anti-HA antibodies and analyzed by the indicated antibodies using WB. (**e–f**) β-tubulin was measured and analyzed as an input control. Three biological repeats of each experiment were repeated independently with similar results. Source data are provided as a Source Data file.

A2, A3, M5, L9, V15, L16, L18, P20 and I21 (Fig. 3a). A stable engagement between HBx$_{2-21}$ and Spindlin1 is further supported by parallel β-sheet formation between "β1$_{(HBx2-21)}$-β4$_{(Tudor 3)}$" and "β2$_{(HBx2-21)}$-β1$_{(Tudor 3)}$" (Fig. 3b, c). Five residues K216, V218, V232, I245 and L258 of Spindlin1 Tudor domain 3 constitute the hydrophobic core with M5, L16 and L18 of HBx (Fig. 3d).

To validate the significance of specific interactions identified in the structure of the Spindlin1-HBx complex, we next generated several Spindlin1$_{50-262}$ mutants and quantitatively evaluated the effects of mutation by ITC titrations (Fig. 3e). As expected, all three mutants were well expressed and highly purified, and no binding affinity was detected when bulky arginine mutations of residues V232 and I245

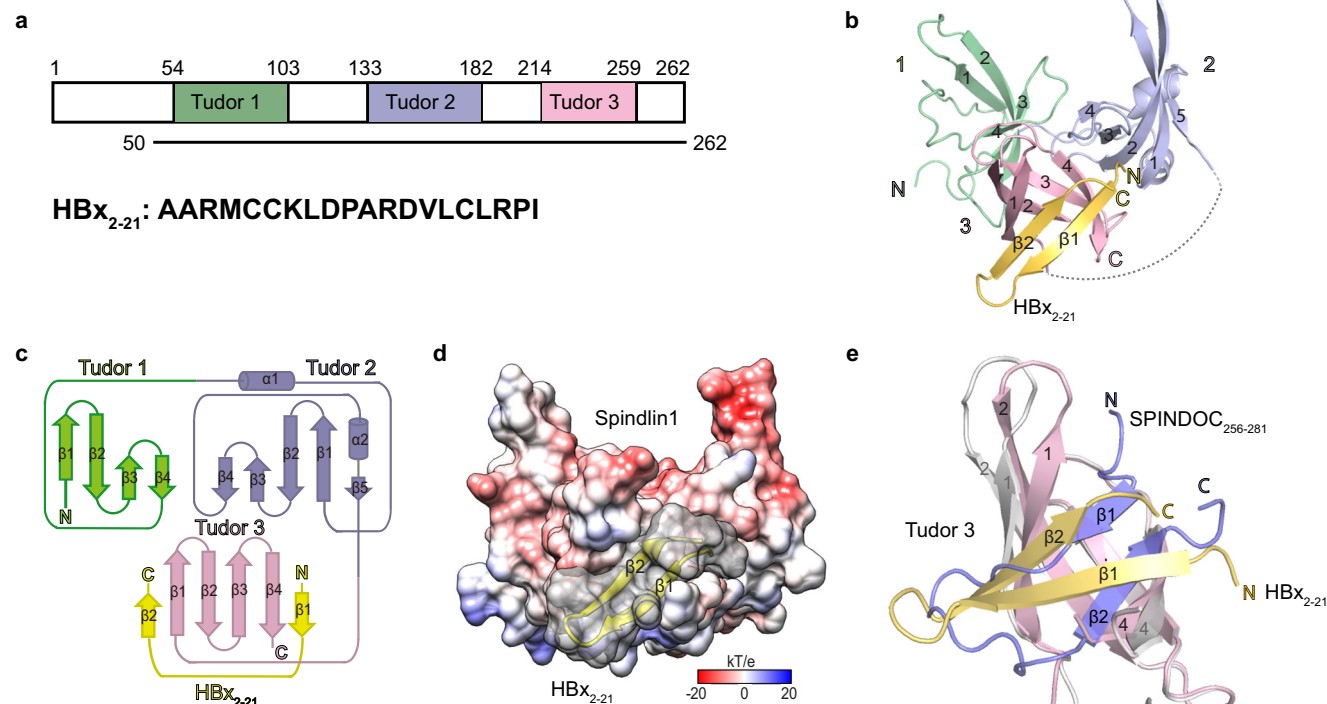

**Fig. 2 | Overall structure of Spindlin1-HBx$_{2-21}$ complex. a** Domain architecture of Spindlin1 (upper) and sequence of HBx$_{2-21}$ (down). Three Core β-barrel regions of the Spindlin1 Tudor repeats are colored green, light blue, and pink, respectively. **b** Overall structure of Spindlin1$_{50-262}$ bound to HBx$_{2-21}$ peptide in ribbon view. HBx$_{2-21}$ is depicted as a yellow ribbon. **c** Folding topology of Spindlin1 and HBx$_{2-21}$. The helices are drawn as barrels, the β-strands are drawn as open arrows, and both are numbered. **d** Electrostatic potential surface view of Spindlin1$_{50-262}$ in complex with HBx$_{2-21}$ peptide. Electrostatic potential is expressed as a spectrum ranging from -20 kT/e (red) to +20 kT/e (blue). **e** Cartoon representation of structural comparison of Spindlin1-HBx$_{2-21}$ complex (pink-yellow) with Spindlin1-SPINDOC$_{256-281}$ complex (PDB ID 7E9M, light gray-slate).

were introduced. Similarly, K216S/V218R double mutant caused a totally binding loss as well. We also tested 4 HBx mutants in the region spanning amino acids 2 to 21 (Fig. 3f). Compared to wild-type peptide, the corresponding mutant HBx$_{2-21}$ (V15R or L18R) peptide resulted in a 14-fold or 36-fold reduction in binding affinity with Spindlin1, respectively. Especially, mutant HBx$_{2-21}$ (V15E/C17R or AAAAA) displayed no detectable binding. Collectively, these results indicate that these hydrophobic residues are important for the interaction between Spindlin1 and HBx.

### HBx engagement is compatible with histone binding by Spindlin1

Spindlin1 is a multifunctional histone methylation reader, which recognizes H3K4me3, H4K20me3, as well as "K4me3-R8me2a" and "K4me3-K9me3/2" methylation patterns of H3 through its first and second Tudors. To explore the interplay between HBx engagement and histone recognition by Spindlin1, we performed ITC experiments using free state and HBx/histone peptides-bound Spindlin1 samples. We found that pre-binding of H3 "K4me3-K9me3" by Spindlin1, as well as other, reported histone marks, such as H3K4me3, H3 "K4me3-R8me2a" and "K4me3-K9me2" and H4K20me3 displayed minimal impact on HBx$_{2-21}$ engagement with less than twofold binding affinity variations (Fig. 4a and Supplementary Fig. 3a). Notably, the engagement of Spindlin1 with HBx$_{2-21}$ exhibits about threefold enhancement in binding to histone H3 "K4me3-K9me3/2" and unchanged affinities towards H3K4me3, H3 "K4me3-R8me2a", and H4K20me3 (Fig. 4b and Supplementary Fig. 3b). To explore the underlying molecular basis, we performed structural alignment of the Spindlin1-HBx$_{2-21}$ complex with reported co-crystal structures of Spindlin1 bound to

different histone peptides. It was shown that HBx engagement by Tudor 3 and histone binding by Tudors 2 and 1 concurrently occur without discernable structural conflicts (Fig. 4c and Supplementary Fig. 3c). Functionally, our RT-qPCR studies in HEK 293 T cells showed that overexpression of Spindlin1 with or without full-length HBx displayed similar transcription-promoting activities towards *CyclinD1*, *Axin2* and *pre-rRNA* genes (Fig. 4d). ChIP-qPCR assays further confirmed that HBx did not affect the enrichment of Spindlin1, histone H3K4me3, and H3K9me3 marks at the promoter regions of *CyclinD1*, *Axin2* and *rDNA genes* (Fig. 4e and Supplementary Fig. 3d, e). Collectively, these studies establish that HBx engagement is compatible with histone readout and the transcriptional co-activator functions of Spindlin1.

### Spindlin1-HBx interaction promotes HBV transcription

Considering the essential role of HBx in HBV replication and transcription, we wondered whether the interaction of Spindlin1 with HBx would affect HBV life cycle in hepatocytes. Firstly, our ChIP experiments in HBV infected HepG2 cells stably expressing NTCP (HepG2-NTCP) indicated that Spindlin1 was enriched at HBV cccDNA minichromosome (Fig. 5a). Then, Northern Blot and RT-qPCR analysis indicated that depletion of Spindlin1 leads to a significant reduction in the HBV transcription levels (Fig. 5b, c and Supplementary Fig. 4a, b), as well as HBV e-antigen (HBeAg, a marker for HBV gene expression) levels in the medium supernatant (Fig. 5d and Supplementary Fig. 4c) and HBV core antigen (HBcAg) levels in the nucleus (Fig. 5e). Moreover, our results in HBV-infected primary human hepatocytes (PHHs) also revealed that knockdown of Spindlin1 caused a significant reduction in the HBV RNA levels and HBeAg levels (Supplementary Fig. 4d–f).

## Table 1 | Data collection and refinement statistics

| | Spindlin1$_{50-262}$-HBx$_{2-21}$ |
|---|---|
| PDB code | 8GTX |
| Data collection | |
| Wavelength (Å) | 0.9792 |
| Space group | P2$_1$2$_1$2$_1$ |
| Cell dimensions | |
| a, b, c (Å) | 47.8, 67.1, 93.7 |
| α, β, γ (°) | 90, 90, 90 |
| Resolution (Å) | 50-1.8 (1.83–1.80) |
| $R_{merge}$ (%) | 10.1 (57.7) |
| $I / \sigma I$ | 17.4 (2.35) |
| Completeness (%) | 99.9 (100.0) |
| Redundancy | 6.4 (6.5) |
| Refinement ($F > 0$) | |
| Resolution (Å) | 46.9-1.80 |
| No. of reflections | 28,688 |
| $R_{work} / R_{free}$ (%) | 16.5/19.7 |
| No. of atoms | |
| Protein | 1618 |
| HBx(2–21) | 153 |
| Water | 302 |
| $B$-factors (Å2) | |
| Protein | 21.8 |
| HBx(2–21) | 25.1 |
| Water | 33.6 |
| R.m.s. deviations | |
| Bond lengths (Å) | 0.006 |
| Bond angles (°) | 0.776 |
| Ramachandran Plot | |
| Favored (%) | 97.66 |
| Allowed (%) | 2.34 |
| Outliers (%) | 0 |

Finally, our rescue results indicated that ectopic expression of wild-type (WT) Spindlin1 can restore the levels of HBV transcription and HBeAg in Spindlin1 knockdown cells (Fig. 5f, g), implying the important role of Spindlin1 in stimulating HBV transcription.

To monitor if the promoted HBV transcription by Spindlin1 is owing to its binding to HBx, we assessed the recruitment of Spindlin1 on the cccDNA in wild-type HBV or HBx-deficient HBV (HBV X-) infected HepG2-NTCP cells. As indicated in Fig. 5h, we detected clear Spindlin1 signal on cccDNA in HepG2-NTCP cells infected with wild-type HBV. By contrast, the enrichment of Spindlin1 was significantly decreased when the HBx-deficient HBV was used for infection, suggesting a role of HBx in stabilizing Spindlin1 at HBV cccDNA minichromosome. Meanwhile, the enrichment of Spindlin1 on cccDNA still exists compared to the anti-IgG control, suggesting that Spindlin1 could target the cccDNA minichromosome via an HBx-independent mechanism. In fact, this is consistent with our observation that Spindlin1 is a potent reader of the histone H3 "K4me3-K9me3" methylation pattern ($K_d$ = 3.8 nM) that marks poised heterochromatic regions (Fig. 4b). Our structural and ITC studies have identified key residues of Spindlin1 Tudor 3 essential for HBx binding. In particular, K216S/V218R double mutation disrupted Spindlin1-HBx engagement both in vitro and in cellular context (Fig. 3e, Supplementary Fig. 4g). Then we did rescue experiments in Spindlin1 knockdown HepG2-NTCP cells by ectopic expression of wild-type and HBx-binding deficient Spindlin1. It was wild-type Spindlin1 but not the K216S/V218R double mutant that effectively restored the levels of HBV transcription and HBeAg (Fig. 5i, j), underscoring the importance of Spindlin1-HBx interaction in promoting HBV transcription. Of note, our Southern Blot analysis confirmed that knockdown of Spindlin1 did not affect cccDNA formation (Supplementary Fig. 4h, i).

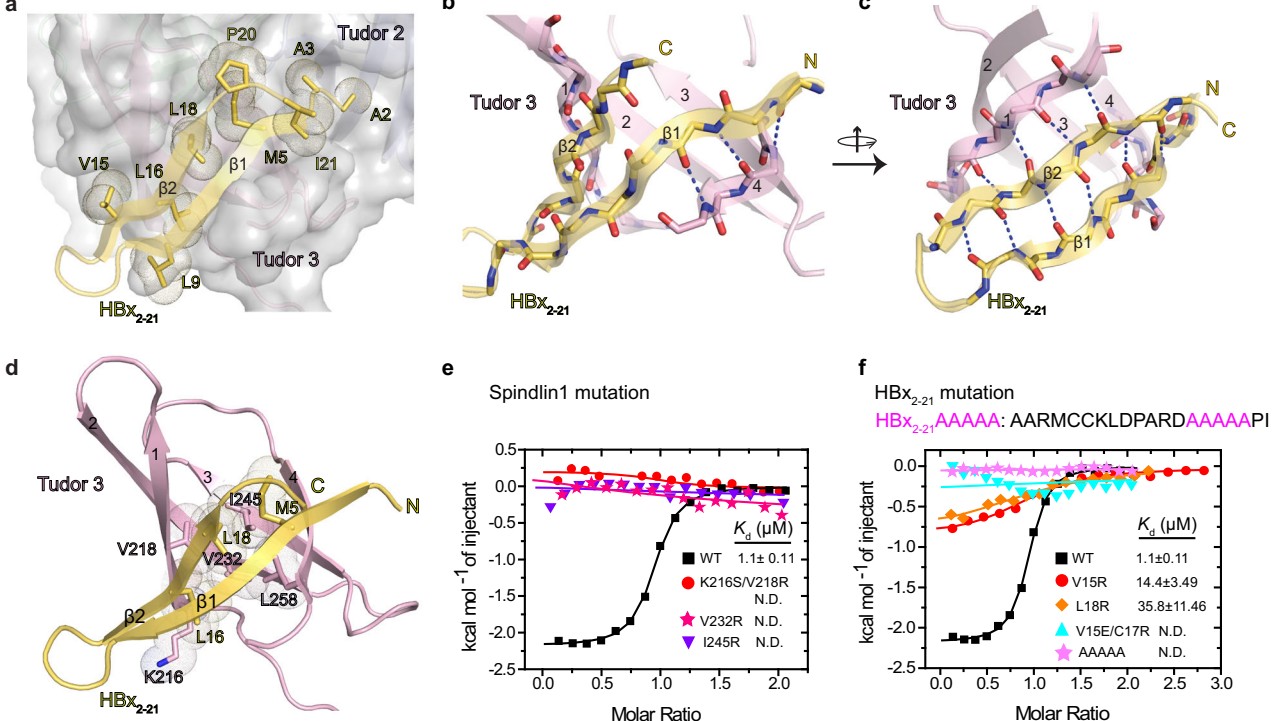

**Fig. 3 | Binding details of Spindlin1-HBx$_{2-21}$ complex and mutagenesis studies.**
**a** Hydrophobic interface between Spindlin1 (surface view in light gray) and HBx$_{2-21}$ (ribbon view in yellow) with hydrophobic residues showing dots. **b** Parallel β-sheet formed by β1 strand (yellow) of HBx$_{2-21}$ and β4 strand (pink) of Spindlin1 Tudor 3. **c** Parallel β-sheet formed between β1 of Spindlin1 Tudor 3 and β2 of HBx$_{2-21}$.

(b-c) Blue dashes: direct hydrogen bonds or salt bridges. **d** Hydrophobic core formed by Spindlin1 K216, V218, V232, I245, L258 (shown as stick in pink) and HBx M5, L16, L18 (shown as stick in yellow). **e** ITC fitting curves of HBx$_{2-21}$ peptide titrated to Spindlin1$_{50-262}$ Tudor 3 mutants. **f** ITC fitting curves of mutant HBx$_{2-21}$ peptides titrated to Spindlin1$_{50-262}$. N.D., not detectable.

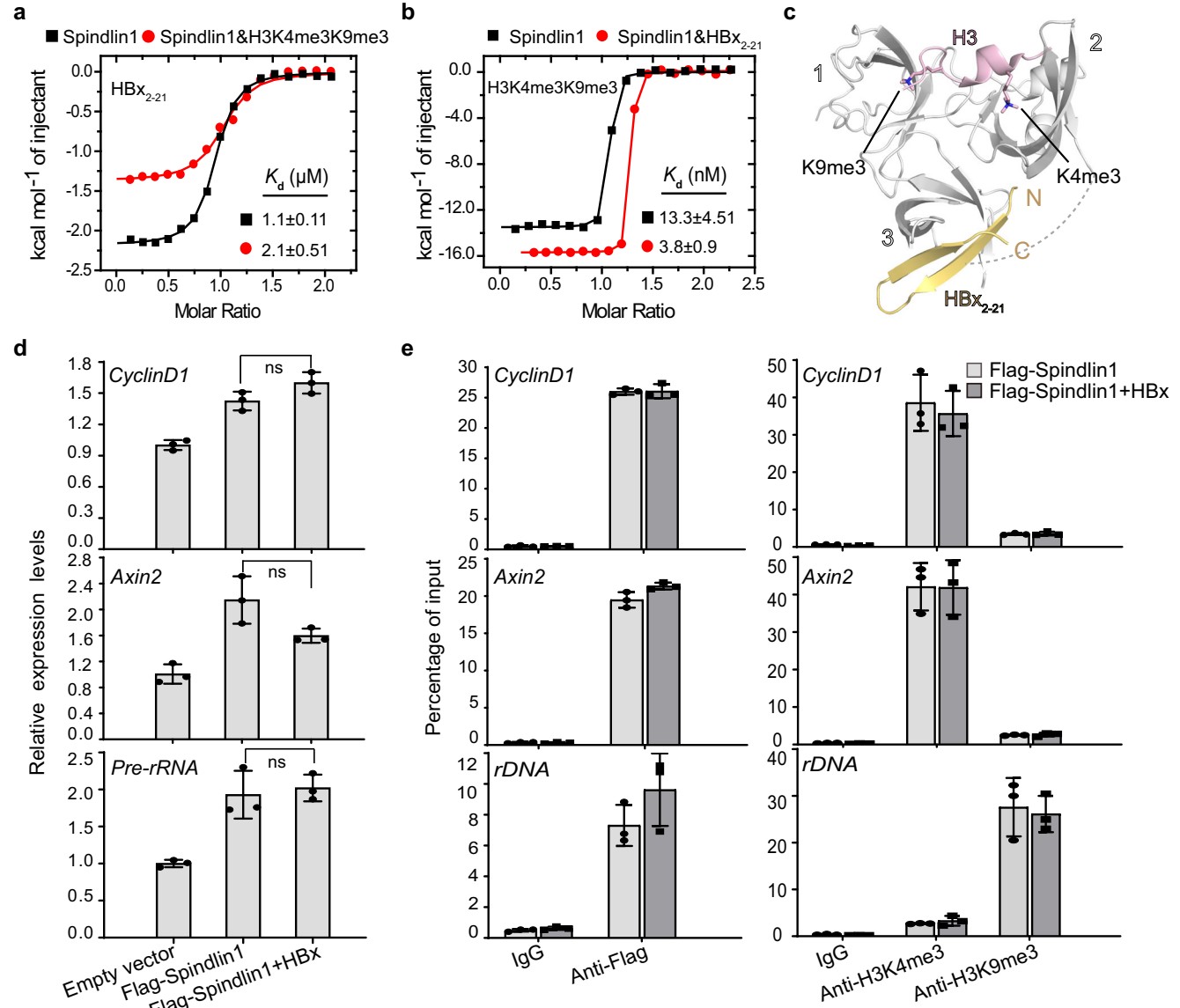

**Fig. 4 | Effect of Spindlin1·HBx engagement on histone readout by Spindlin1.**
**a** ITC fitting curves of HBx$_{2-21}$ peptide titrated to free Spindlin1$_{50-262}$ or Spindlin1$_{50-262}$ incubated with H3 "K4me3-K9me3" peptide. **b** ITC fitting curves of H3 "K4me3-K9me3" peptide titrated to free Spindlin1$_{50-262}$ or Spindlin1$_{50-262}$ incubated with HBx$_{2-21}$ peptide. **c** Cartoon representation of structural comparison of Spindlin1$_{50-262}$ bound to HBx$_{2-21}$ with Spindlin1$_{50-262}$ bound to H3 "K4me3-K9me3" (PDB ID 7BQZ). **d** RT-qPCR analysis of *CyclinD1*, *Axin2* and *Pre-rRNA* genes

in HEK 293 T cells transfected with empty vector, Spindlin1 or Spindlin1 plus HBx. **e** Enrichment of Spindlin1 and histone H3K4me3 and H3K9me3 marks at the promoter regions of *CyclinD1*, *Axin2*, and *rDNA genes* by ChIP-qPCR in HEK 293 T cells, which were transfected with Flag-Spindlin1, or Flag-Spindlin1 plus HBx. **d**–**e** Data represent the mean ± SD ($n = 3$ independent experiments). ns, not significant difference (unpaired $t$ test, two-tailed). Source data are provided as a Source Data file.

## Spindlin1·HBx regulates HBV transcription through cccDNA chromatin modifications

Considering that histone methylation readout by Spindlin1 is required for transcriptional activation[29,30], we wondered whether the interplay between Spindlin1 and HBx promotes HBV transcription through cccDNA chromatin modifications. First, we examined the histone H3 methylation states of cccDNA by ChIP-qPCR using antibodies against histone H3K4me3 and H3K9me3 in HepG2-NTCP cells. We showed that the H3K4me3 mark was enriched at cccDNA in cells infected with wild-type HBV, while the repressive H3K9me3 mark was enriched at cccDNA in cells infected with HBx-deficient HBV (Fig. 6a, b). Consistent with a role of "Spindlin1-H3K4me3" recognition pair in active transcription, we observed clear bands of HBV transcripts by Northern Blot in wild-type but not in HBx-deficient HBV infected cells (Fig. 6c).

Next, we knocked down Spindlin1 in HepG2-NTCP cells infected with wild-type HBV (Fig. 6d). To our expectation, H3K4me3 enrichment on HBV cccDNA was remarkably decreased, whereas H3K9me3 levels were significantly increased (Fig. 6e, f). We also explored the genomic rDNA locus, and observed similar decrease of H3K4me3 upon Spindlin1 knockdown although the changes of H3K9me3 level is minimal (Fig. 6g, h). Collectively, these data suggest that Spindlin1·HBx engagement promotes HBV transcription associated with upregulated H3K4me3 levels on chromatinized HBV cccDNA. In support, ectopic expression of wild-type Spindlin1 but not its H3K4me3 and H3K9me3-binding deficient mutants, such as W72R, F141A, Y170A and W72A/F141A, restored HBV expression in Spindlin1 knockdown cells (Fig. 5i, j), suggesting methylated histone readout is required for Spindlin1 to effectively activate HBV transcription.

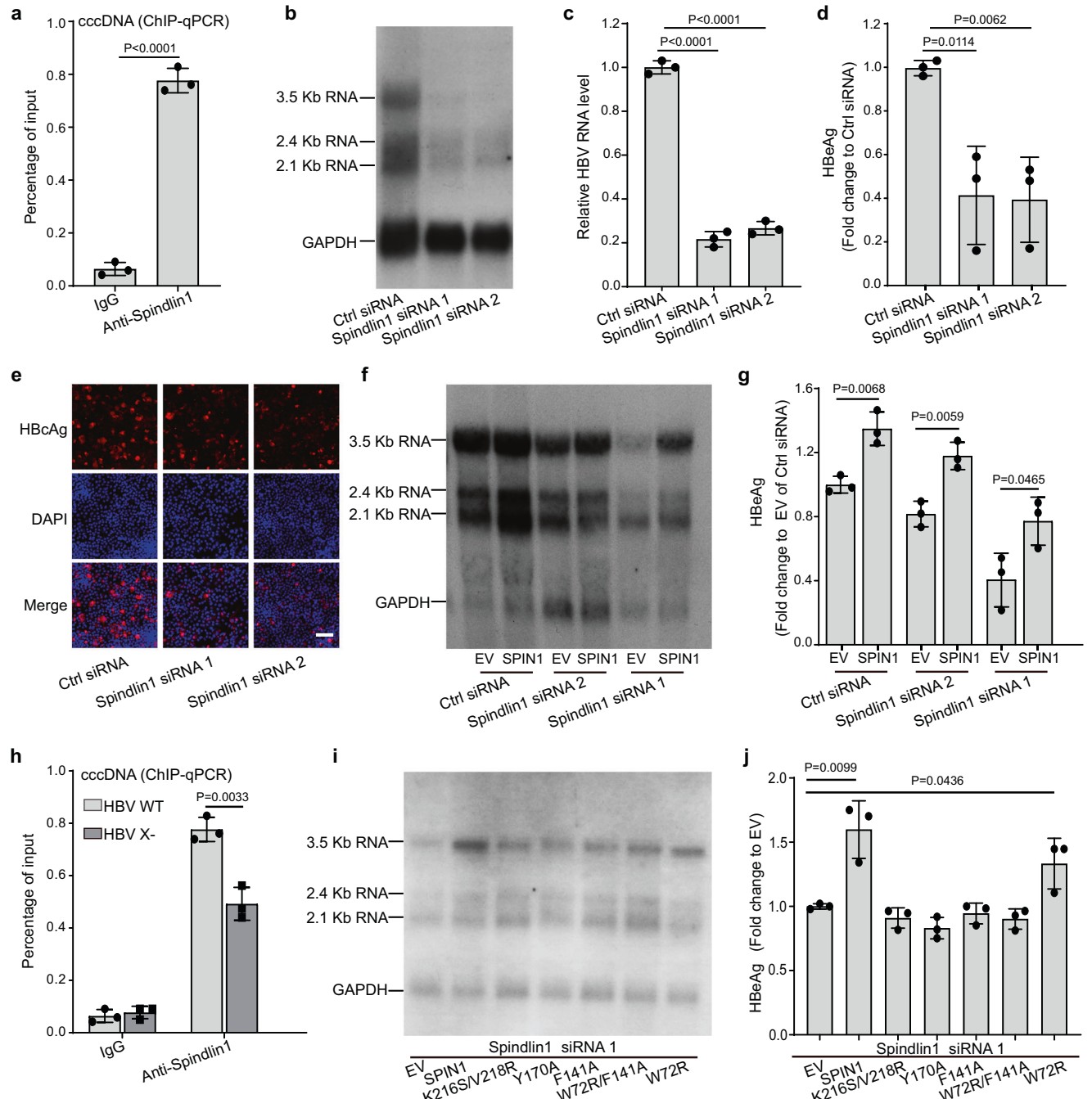

**Fig. 5 | Spindlin1-HBx promotes HBV gene transcription in HepG2-NTCP cells.**
**a** ChIP experiments using HBV infected cells showed Spindlin1 is recruited to the HBV cccDNA. **b–e** At 3 days post infection (dpi) with HBV, cells were transfected with control siRNA (Ctrl siRNA) or Spindlin1 siRNAs and collected at 10dpi for HBV RNA (**b, c**), HBeAg (**d**), and HBcAg (red, **e**) levels analysis. **e** Cell nuclei were stained with DAPI dye (blue). Scale bars: 100 μm. Three biological repeats of each experiment were repeated independently with similar results. **f–g** Ectopic expression of wild-type Spindlin1 (SPIN1) restored the HBV transcription (**f**) and HBeAg (**g**) levels in Spindlin1 knockdown cells. **h** HepG2-NTCP cells were infected with normalized amount of wild-type HBV (HBV WT) or HBx-deficient HBV (HBV X-). At 7dpi, cells were harvested and analyzed by ChIP using antibodies against Spindlin1. **i–j** Ectopic expression of wild-type Spindlin1 but not its H3K9me3 or H3K4me3 or double-binding deficient mutant restored HBV expression (**i**) and HBeAg (**j**) levels in Spindlin1 knockdown cells. EV, empty vector. Data represent the mean ± SD ($n = 3$ independent experiments). *P*-values between the groups were calculated with an unpaired two-tailed *t* test. Source data are provided as a Source Data file.

## Crosstalk between Spindlin1-HBx and DDB1-HBx engagements

DDB1 is one of the well-known partners of HBx in regulating HBV cccDNA transcription, to study whether Spindlin1-HBx binding is related to DDB1-HBx interaction, we analyzed the structure of HBx protein. The predicted structure of HBx by AlphaFold[40] shows that elements HBx$_{2–21}$, HBx$_{88–100}$ and HBx$_{104–135}$ cluster to form a hydrophobic core (Fig. 7a). Interestingly, in the folded HBx structure, key

residues involved in Spindlin1, DDB1 and Bcl-2 binding, such as M5, L16, L18 of HBx$_{2–21}$, L89, L93, R96 of HBx$_{88–100}$ and V116, W120, L123, I127, R128, F132 of HBx$_{110–135}$, are buried upon hydrophobic core formation. This suggests that effective engagement of HBx with DDB1 and Spindlin1 requires unfolding of HBx to expose key binding motifs. In this case, we speculate that HBx may exist in two conformational states: a folded inactive state and an extended active state (Fig. 7b).

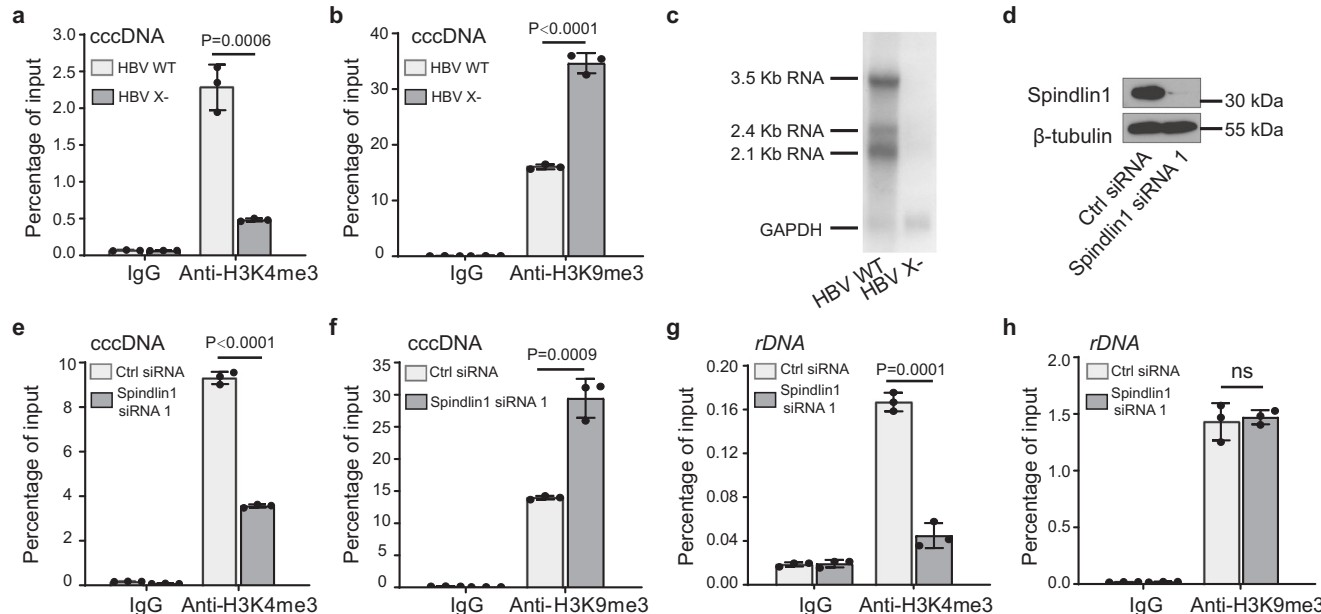

**Fig. 6 | Spindlin1-HBx regulates HBV transcription with reprogrammed cccDNA chromatin modifications. a–c** HepG2-NTCP cells were infected with normalized amount of HBV WT or HBV X-. At 7dpi, cells were harvested and analyzed by ChIP using antibodies against H3K4me3 (**a**) and H3K9me3 (**b**), and HBV RNA levels were quantified by Northern blot (**c**). **d–h** Effect of depletion of Spindlin1 (**d**) on enrichment of H3K4me3 and H3K9me3 at HBV cccDNA (**e, f**) and *rDNA* (**g, h**) in HBV infected HepG2-NTCP cells. (**a–b, e–h**) Input and immunoprecipitated DNA were analyzed in triplicate by qPCR with primers specific for cccDNA or *rDNA* and were displayed as percentage of input. As a control, immunoprecipitation was performed with rabbit IgG. Data represent the mean ± SD ($n = 3$ independent experiments). *P*-values between the groups were calculated with an unpaired two-tailed *t* test. ns, not significant difference. Source data are provided as a Source Data file.

Conceivably, Spindlin1, DDB1 and Bcl-2 may cooperate with each other to interact with HBx by jointly overcoming the folding energy barrier, thus enabling a functional switch of HBx from an inactive state to an active one.

To experimentally investigate the functional correlation between Spindlin1 and DDB1, we conducted HBx pulldown and SMC6 degradation assays in HepG2-NTCP cells. We showed that siRNA knockdown of Spindlin1 resulted in slightly decreased level of DDB1 that binds to HBx (Fig. 7c). In SMC6 degradation assays, our western blot analysis revealed a significant decrease in the SMC6 level in cells expressing HBx. Remarkably, such a decrease was partly restored in Spindlin1 knockdown cells (Fig. 7d). These observations are consistent with our structural analysis above, and support a positive crosstalk between Spindlin1-HBx and DDB1-HBx engagements as well as DDB1-HBx-mediated SMC5/6 degradation.

## Discussion

Spindlin1 has been characterized as a transcriptional coactivator through histone methylation readout involving the first and second Tudor domains. Here we characterized a unique engagement mode between HBx and the third Tudor of Spindlin1. Our binding, structural and functional studies suggested that the molecular functions of all three Tudors are coordinated to sustain a transcriptional coactivator function of Spindlin1 towards the cccDNA minichromosome. We demonstrated that Spindlin1 interacts with HBx and concurrently recognizes histone H3K4me3 or bivalent histone H3 "K4me3-K9me3" methylation pattern. Functionally, Spindlin1-HBx engagement plays a role in promoting HBV transcription, associated with chromatin modification switch from H3K9me3-enriched repressive to H3K4me3-marked active state, as well as a conformational switch of HBx that may occur in coordination with other HBx-binding factors.

In previous studies, Spindlin1 was characterized as a host restriction factor of HBV in HepAD38 cells and was proposed to act as an intrinsic antiviral defense mechanism against DNA viruses[24]. However, our studies in HepG2-NTCP cells suggested that Spindlin1 promotes HBV transcription in an HBx-engagement dependent manner. The observed opposite roles of Spindlin1 in HBV transcription might attribute to different research systems used and reflect the regulatory complexity of epigenetic "ON-OFF" control. Our knockdown studies showed that Spindlin1 is required for efficient HBV transcription from the chromatinized cccDNA (Fig. 5). Moreover, our rescue experiments showed that ectopic expression of wild-type but not Tudor function deficient Spindlin1 could rescue HBV transcription in Spindlin1 knockdown HepG2-NTCP cells. From the perspective of molecular evolution, the fact that HBV exploits a highly conserved N-terminal motif of HBx to interact with Spindlin1 also suggests a proviral rather than an anti-viral role of Spindlin1-HBx engagement in HBV life cycle. Otherwise, mutations in HBx should have occurred to HBV to escape Spindlin1 restriction.

Once entering the cell nucleus, the DNA of many viruses are often epigenetically silenced by host cells through establishment of H3K9me3-marked heterochromatin in association with subnuclear bodies like ND10/PML-NBs[15]. The formation of cccDNA and its chromatinization are crucial for HBV chronic infection. To overcome epigenetic repression, dedicated mechanisms are required to establish an active chromatin state of HBV cccDNA. We reasoned that the cccDNA minichromosome of HBV may exist in three chromatin states: an H3K9me3-enriched repressive state, a bivalent H3 "K4me3-K9me3"-marked poised state, and an H3K4me3-marked active state (Fig. 7e). Remarkably, the Spindlin1-HBx$_{2-21}$ complex displayed about 12-fold binding enhancement towards H3 "K4me3-K9me3" peptide ($K_d$, 3.8 nM) as compared to H3K4me3 ($K_d$, 44.8 nM) (Fig. 4b and Supplementary Fig. 3b). Thus, the Spindlin1-HBx complex likely serves as a delicately evolved effector for the poised state cccDNA minichromosome, and functions to trigger an epigenetic shift towards H3K4me3-marked active chromatin. It has been reported that the SET1/MLL complexes can methylate H3K4 in the presence of H3K9me3[49]. Therefore, H3K4me3 can be enzymatically created in H3K9me3-rich regions to establish a bivalent H3 "K4me3-K9me3" methylation pattern for Spindlin1 recruitment. On the other hand, it has been well

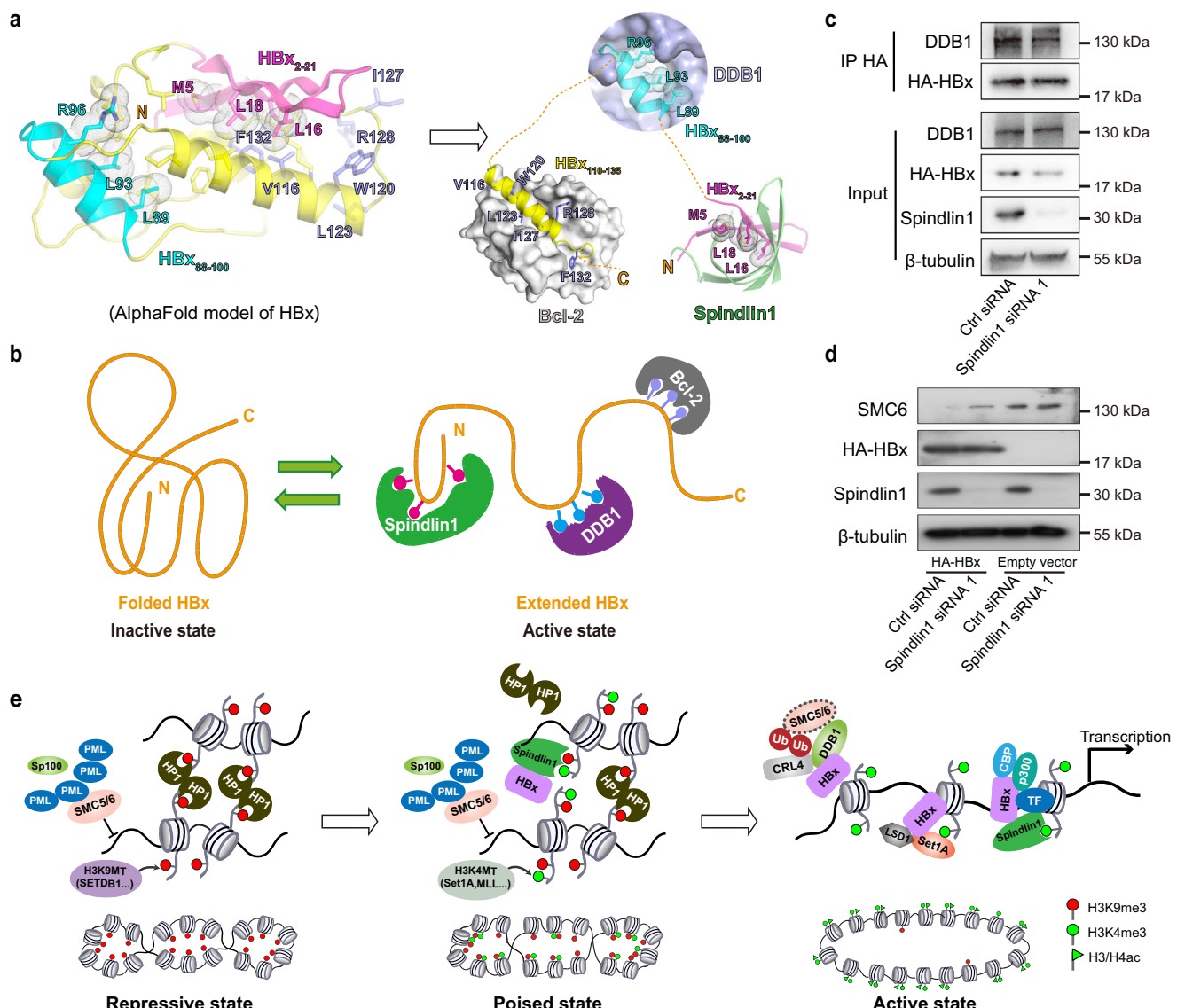

**Fig. 7 | Crosstalk between Spindlin1-HBx and DDB1-HBx engagements.**
**a** AlphaFold model of free HBx (left) and structure model of HBx binding to Spindlin1 (light magenta-green), DDB1 (cyan-gray, PDB ID 3I7H), and Bcl-2 (yellow-white, PDB ID 5FCG). **b** Schematic model of HBx transitioning from an inactive state to an active one by binding to Spindlin1, DDB1, and Bcl-2. **c** Co-IP of HA-HBx with endogenous DDB1 using anti-HA antibodies in HepG2-NTCP cells, which were transfected with Ctrl siRNA or Spindlin1 siRNA 1. **d** WB analyzing the SMC6 level in Spindlin1 knockdown HepG2-NTCP cells expressing HBx. Cells were transfected with Ctrl siRNA or Spindlin1 siRNA 1 for 72 h and then transfected with HA-HBx or empty vector. After 96 h, cells were harvest for WB analyzation. (**c**–**d**) $n$ = 3 independent experiments. **e** A schematic model of Spindlin1-HBx mediated transcriptional derepression of cccDNA minichromosome. Chromatinized HBV genome

undergoes epigenetic reprogramming from a repressive state to an active state via a poised intermediate state. The transcriptionally inactive HBV genome is typically silenced by host restriction factors, such as PML, Sp100, SMC5/6, H3K9 methyltransferases (H3K9MT), and HP1, which form a repressive chromatin state marked by H3K9me3 (left). To overcome the heterochromatin barrier, histone H3K4 methyltransferases (H3K4MT) can target H3K9me3-containing cccDNA chromatin to create bivalent H3 "K4me3-K9me3" modifications[49]. Spindlin1-HBx complex then binds to H3 "K4me3-K9me3" and competes off HP1 from the poised state chromatin (middle). Finally, Spindlin1 and HBx cooperatively establish an active chromatin state of HBV cccDNA in concert with transcription factor (TF) and co-activators such as CBP/p300 to promote HBV transcription (right). Source data are provided as a Source Data file.

established that erasers of H3K9me2/3, such as KDM4 and KDM7 family members, can recognize H3K4me3 to remove H3K9me2/3 marks. This "read-erase" mechanism may function to resolve the H3 "K4me3-K9me3" bivalent state and establish a fully active chromatin state marked by H3K4me3 and hyperacetylation. Further work will be needed to illustrate the detailed molecular mechanisms underlying Spindlin1-HBx-mediated epigenetic reprogramming of the cccDNA minichromosome.

Noteworthily, this is not the first time that Spindlin1 was reported to promote gene expression and chromatin state transition through recognition of H3 "K4me3-K9me3" in concert with

partner engagement via its third Tudor domain. Previous studies showed that combinatorial readout of H3 "K4me3-K9me3" by Spindlin1 in complex with SPINDOC could displace HP1 proteins from the poised rDNA loci, thereby facilitating rRNA expression[29,30]. H3K9me3-mediated heterochromatinization is an important strategy to silence HBV genome and rDNA repeats[20,41]. Intriguingly, to turn on gene expression, Spindlin1 is exploited by both host cell and virus to overcome the heterochromatin barrier as a potent H3K4me3 reader in H3K9me3-enriched regions. Meanwhile, it has not escaped our attention that HBx and SPINDOC adopt rather different engagement modes with Tudor 3 of Spindlin1 (Fig. 2e,

Supplementary Fig. 2b, c). On the one hand, it highlights a role of Tudor 3 in diverse non-histone partner engagement; on the other hand, different engagement modes may lead to different outcomes in gene regulation. We have previously shown that although SPINDOC-Spindlin1 engagement does not affect H3 "K4me3-K9me3" readout by Spindlin1, it significantly weakens H3K4me3 or TCF4 binding, therefore inhibiting transcription of H3K4me3/TCF4-driven genes and facilitating the redistribution of Spindlin1 to H3 "K4me3-K9me3"-enriched poised regions[29]. In the case of HBx, we observed unaffected or enhanced histone binding by Spindlin1 due to its unique engagement mode, well consistent with the transcription promoting role of Spindlin1-HBx towards HBV cccDNA minichromosome.

As the sole HBV regulatory protein, HBx is well conserved among mammalian hepadnaviral family members. Alignment of different human HBx subtypes as well as mammalian HBx proteins showed that the N-terminal motifs ($HBx_{2-21}$) are of high sequence homology, especially the key residues involved in Spindlin1 engagement (Supplementary Fig. 5a). Our ITC assays confirmed micromolar level binding between these motifs and Spindlin1 (Supplementary Fig. 5b), underscoring a conserved function of Spindlin1 in promoting viral gene expression among a wide range of hepadnaviral family members. Interestingly, the N-terminal motif of HBx has been implicated a transrepressor function by inhibiting the transactivation activity of the C-terminal region ($HBx_{51-154}$)[42]. However, here we revealed a transcriptional activation role of $HBx_{2-21}$ by recruiting Spindlin1. This seemingly contradictory roles of $HBx_{2-21}$ may be reconciled by the conformational switch of HBx state triggered by Spindlin1. HBx has been shown to interact with both transcriptional co-activators, such as DDB1, CBP/p300, GCN5, SET1A, and transcriptional co-repressors, such as DNMT3A, HP1, HDAC1, SETDB1[6,20,21,43]. Conceivably, in the absence of Spindlin1, $HBx_{2-21}$, $HBx_{88-100}$ and $HBx_{104-135}$ cluster likely form a stable hydrophobic core, and represses the transactivation activity of HBx by burial of DDB1 or Bcl-2 binding motifs (Fig.7a, b). Whereas, the engagement of $HBx_{2-21}$ with Spindlin1 likely unlocks the transrepression activity of the N-terminal motif, and enables a functional switch of HBx from organizing a repressive chromatin state to an open one by cooperatively recruiting activation-related factors such as DDB1, CBP/p300, or other HBV-specific transcription factors.

HBV infection still exists as a major health problem worldwide. Here our structural, biochemical, and virological studies established a critical function of Spindlin1-HBx interplay in overcoming the epigenetic barrier imposed on HBV cccDNA by host restriction factors. As interference of Spindlin1 function by RNAi impaired HBV replication, our work not only illustrates the regulatory complexity in HBV life cycle, but also paves the way for new therapeutic opportunities by targeting Spindlin1-HBx to harness chronic infection of HBV and the treatment of HBV-induced liver diseases.

## Methods
### Protein and peptide preparation
Human $Spindlin1_{50-262}$ was cloned into pGEX-6P-1 vector containing an N-terminal GST tag and overexpressed in *E. coli* strain BL21(DE3) at 16 °C for 18 h and induced by 0.2 mM isopropyl-β-D-thiogalactopyranoside (IPTG). Cells were harvested and lysed in lysis buffer: 0.2 M NaCl, 20 mM Tris, pH 8.0 containing 1 mM fresh PMSF. After centrifugation, the GST-tagged $Spindlin1_{50-262}$ was purified by a GST affinity column and then the GST tag was cleaved on-column by a home-made P3C protease overnight. The tag-free protein was collected for an anion-exchange chromatography over a HiTrap Q HP column and Superdex G75 columns (GE Healthcare). The $Spindlin1_{50-262}$ protein was concentrated to 25 mg/ml in 0.15 M NaCl, 20 mM Tris, pH 8.0, aliquoted and stored at −80 °C for future use. All $Spindlin1_{50-262}$ mutants were generated by the QuikChange site-

directed mutagenesis strategy and purified using the same procedure as described above.

HBx protein (adw4 subtype) was chopped into 14 peptides with a 10 amino acid residues-overlap between the adjacent peptides. All synthetic HBx peptides and histone H3 peptides (>95% purity) were purchased from Scilight Biotechnology LLC and described in the Supplementary Table 1.

### Surface plasmon resonance imaging (SPRi) assay
To profile the interaction between HBx peptides and Spindlin1 protein in a high-throughput manner, an SPRi instrument (Kx5, Plexera, USA) was used to monitor the whole procedure in real-time according to the previously reported method[44]. In brief, the gold surface of the SPRi chip was chemically modified with diazirine-terminated polymers. The HBx peptides were prepared in a 384-well plate at 1 mM in $ddH_2O$. After microarray printing using a robotic microarrayer (A3210, Biodot, USA), the HBx peptides were immobilized onto the SPRi chip surface via UV photo-crosslinking. Then, 1 μM of human $Spindlin1_{50-262}$ protein in running buffer was applied to the chip surface with 500 s association and 500 s dissociation. Finally, the kinetic binding curves were extracted from the imaging data, and the signals were converted to standard refractive units (RU) through calibrating every spot with 1% glycerol (w/v) in running buffer with known refractive index change (1200 RU). The standard errors of binding parameters are calculated by fitting all data points by a commercial software (Plexera SPR Data Analysis Module, Plexera, USA).

### Thermal shift assay (TSA)
The TSA was performed with a CFX96™ real-time PCR instrument (Bio-Rad). A typical TSA solution is composed of 50 μM Spindlin1, 5× Sypro Orange (Invitrogen) in the absence/presence of series of 500 μM HBx peptides. All solutions were prepared in 25 μL under the buffer (150 mM NaCl, 20 mM HepesNa, pH 7.5), with the identity of HBx peptide being the sole variable. During TSA assays, all samples were heated from 25 to 90 °C at a rate of 0.5 °C per minute. Protein denaturation was monitored by increased fluorescence signal of Sypro Orange, which captures exposed hydrophobic residues during thermal unfolding. The recorded curves were analyzed by the software CFX-Manager 3.1 (Bio-Rad). The temperature corresponding to the inflection point was defined as Tm, which means that 50% of the protein is denatured.

### Isothermal titration calorimetry (ITC)
Calorimetric experiments were conducted at 25 °C with a MicroCal PEAQ-ITC instrument (Malvern Instruments) as previously described[29]. The $Spindlin1_{50-262}$ samples were extensively dialyzed against the buffer containing 20 mM HepesNa (pH 7.5) and 150 mM NaCl. Protein concentration was determined by absorbance spectroscopy at 280 nm. Peptide concentrations were determined by weighing in large quantities. Each ITC titration consisted of 17 successive injections with 0.4 μl for the first and 2.4 μl for the rest. The intervals between injections were 180 s, and the stirring speed was 750 rpm. Usually, H3 peptides at 0.25–0.5 mM were titrated into proteins at 0.025–0.05 mM, and HBx peptides at 1 mM were titrated into proteins at 0.1 mM. Acquired calorimetric titration data were fitted at 95% confidence and analyzed using Origin 7.0 (GE Healthcare) using the One Set of Binding Sites fitting model. Data represent the mean ± SE. Detailed thermodynamic parameters of each titration are summarized in the Supplementary Table 2.

### Crystallization and structural determination
$Spindlin1_{50-262}$ was incubated with $HBx_{2-21}$ peptide in a molar ratio of 1:3 for 3 h at 4 °C. Complex crystals were grown at 18 °C by mixing 0.2 μL of protein complex with 0.2 μL of reservoir solution using the sitting drop vapor diffusion method. The $Spindlin1-HBx_{2-21}$ crystal was

grown in buffer consisting of 40 mM Potassium phosphate dibasic, 20% (v/v) Glycerol, 16% (w/v) PEG 8000.

Crystals were flash-frozen in liquid nitrogen under cryoprotectant conditions (reservoir solution supplemented with 10% glycerol). Diffraction data were collected at beamline BL18U of the Shanghai Synchrotron Radiation Facility (SSRF). All data sets were indexed, integrated and scaled with the HKL2000 suite[45]. Crystal structures were determined by molecular replacement using MOLREP[46] in CCP4 with previous published Spindlin1 structure (PDB code: 4MZG) as the search model. Model building and refinement were performed with COOT[47] and PHENIX[48], respectively. Data processing and refinement statistics are summarized in Table 1. Ramachandran plot analysis showed that all residues of Spindlin1-HBx$_{2-21}$ complex structures are within the most favored or allowed regions. Structural analysis and figure preparation were mostly performed using PyMol (http://www.pymol.org).

### Western blot and co-immunoprecipitation
Total proteins were extracted in cell lysis buffer for Western and IP (Beyotime Biotechnology), supplemented with protease inhibitor cocktail and PMSF, and protein concentration was measured by the BCA protein assay kit as recommended by the manufacturer (Beyotime Biotechnology). Western Blot (WB) was performed as follows: proteins were separated with an SDS-PAGE and transferred onto the PVDF membrane. After transfer, the membrane was blocked with 5% skim milk for at least 1 h at room temperature with shaking. The membrane was incubated with the appropriate antibodies overnight at 4 °C with shaking. The next day, the membrane was washed with TBST and incubated with second antibody at room temperature for 1 h. ECL prime western blotting detection reagent (GE) was used to induce chemiluminescence. For co-immunoprecipitation, cell lysates were immunoprecipitated with anti-Flag M2 agarose beads (Sigma-Aldrich) or corresponding antibodies conjugated with protein G beads (Millipore) overnight at 4 °C. Then, the beads were washed for five times with cell lysis buffer, and the bound proteins were eluted in SDS buffer and analyzed by WB.

### Quantitative RT-PCR
Total RNA was extracted from cells using Trizol reagent (Invitrogen) and reversed to cDNA using TransScript® One-Step gDNA Removal and cDNA Synthesis SuperMix kit (AT311). RT-qPCR was performed using a 2×EasyTaq® PCR SuperMix (AS111-13) according to the manufacturer's instructions by a Bio-Rad sequence detection system. Expression data were normalized to the amount of *β-actin*. Relative transcriptional folds were calculated as $2^{-\Delta\Delta Ct}$. Each experiment was repeated independently at least three times. Primer sequences used in RT-qPCR are listed in the Supplementary Table 3.

### Cell culture, plasmids, transfection, viral transduction
HEK 293 T cells (Cat# CRL-3216) were maintained in Dulbecco's modified Eagle's medium (DMEM, Gibco) with 10% FBS. HepG2 cells (Cat# HB8065) were purchased from the American Type Culture Collection (ATCC) and maintained in Dulbecco's modified Eagle's medium (DMEM, Gibco) with 10% FBS, 1xGlutaMAX (Gibco). Human hepatocellular cell (Huh-7, Cat# 3111C0001CCC000679) was obtained from the Cell Bank of Type Culture Collection, Chinese Academy of Sciences and maintained in Dulbecco's modified Eagle's medium (DMEM, Gibco) with 10% FBS, 1xGlutaMAX (Gibco). PHHs (Cat# 00995) were purchased from Shanghai RILD Inc. All cells were maintained at 37 °C and 5% CO$_2$ atmosphere. All culture media were supplemented with penicillin and streptomycin.

The DNA fragment encoding N-terminal HA tagged HBx was amplified from the cDNA of HepAD38 cells by PCR and subcloned into the pcDNA3.1 expression vector. Flag-Spindlin1 was generated by subcloning human Spindlin1 cDNA into the pcDNA3.1 or PLVX-T2A-puro vector. Mutations in full-length Spindlin1 were generated by site-directed mutagenesis and confirmed by DNA sequencing. Plasmid DNA and siRNA transfections were performed using Lipofectamine 3000 reagent (Invitrogen) and Lipofectamine RNAiMAX (Invitrogen) according to the manufacturer's protocol, respectively.

Human lentiviral were produced by HEK 293 T cells. Briefly, HEK 293 T cells were co-transfected with pMD2.G, psPAX2 (Addgene) and PLVX-T2A-puro constructs. For infections, cells were incubated with viral supernatants in the presence of 10 μg/ml polybrene.

### HBV production and infection
HBV genotype D virus was produced by transient transfection of Huh-7 cells with a plasmid containing 1.05 copies of HBV genome under the control of a CMV promoter[3]. HBx protein deficient virus (HBV X-) was generated by co-transfection of Huh-7 cells with a plasmid harboring 1.05 copies of HBV genome with a stop codon at the 8$^{th}$ codon of X gene open reading frame, and an intact HBx protein expression vector. HBV infection assay has been described previously[3]. Briefly, HepG2-NTCP cells were firstly cultured in collagen-coated dish with DMEM complete medium for 3–4 h, and then cells were cultured with PMM medium for another 20 h. The cells were then infected with HBV at 800GEQ in the presence of 5% PEG8000 for 24 h at 37 °C.

For siRNA knockdown experiments of HepG2-NTCP cells, the siRNAs (sequences are listed in the Supplementary Table 4) were transfected at 3dpi, and incubated with cells for 24 h. Then the cells were maintained in PMM with 2% FBS with regular medium changing every other day. For cccDNA formation experiment, the siRNAs were transfected 48 h before infection and incubated with cells for 24 h. Then the cells were cultured with PMM medium for another 20 h. The cells were then infected with HBV at 800GEQ in the presence of 5% PEG8000 for 24 h at 37 °C.

For siRNA knockdown and HBV infection experiments of PHHs, after cell thawing, PHHs were suspended using INVITROGRO CP Medium (Product No. S03316) with 10% FBS and seeded to collagen-coated plate for 6 h. Then PHHs were supplemented with PMM and siRNA was transfected. After incubation for 16 h, PHHs were inoculated with HBV.

### Northern blot analysis of HBV RNA
Cells were harvested with TRIzol™ Reagent (Invitrogen, Carlsbad, CA). The RNA was extracted and precipitated with equal volumes of isopropanol. The RNA pellet was washed with 75% ethanol and dissolved in DEPC-treated H$_2$O. The RNA solution with 1xRNA loading buffer was incubated at 65 °C heat block for 5 min and put on ice for 5 min. Then the RNA sample was subjected to 1.2% agarose gel electrophoresis (with 1xMOPS buffer and 5% formaldehyde solution) and transferred onto Amersham Hybond-N+ membrane (GE Healthcare). After UV crosslinking, the RNA was detected with DIG-labeled full-length HBV genome and DIG-labeled GAPDH probe (DIG-high prime DNA-labeling and detection starter kit II, Roche, cat. No. 11585614910).

### Southern blot analysis of HBV cccDNA
Selective extraction of HBV cccDNA from HBV infected cells was achieved by a modified Hirt method. Briefly, infected cells from one well of 6-well plates were lysed in Hirt lysis buffer (10 mM Tris-HCl, 10 mM EDTA, 0.6% SDS, pH = 7.4) for 2 h at room temperature. After adding 5 M NaCl, the cell lysate was vigorously mixed and incubated at 4 °C overnight. After centrifugation at 13,800 g for 30 min at 4 °C, the supernatant was extracted twice with DNA extraction solution (Solarbio). The extracted DNA was precipitated with equal volumes of isopropanol. The DNA pellet was washed with 75% ethanol and dissolved in TE buffer (10 mM Tris-HCl, 1 mM EDTA, pH = 8.0). Then the extracted cccDNA sample was subjected to 1% agarose gel electrophoresis and transferred onto Amersham Hybond-N+ membrane (GE

Healthcare). After UV crosslinking, the DNA was detected with DIG-labeled full-length HBV genome (DIG-high prime DNA-labeling and detection starter kit II, Roche, cat. No. 11585614910).

## Measurement HBeAg levels
HBeAg levels of HBV infected cells were measured using ELISA kits (Wantai Pharm Inc). Supernatants from HBV infected cells were harvested at each time point examined in the various assays and were diluted 4-fold with DMEM before ELISA.

## Immunofluorescence
Virus infected cells in 48-well plates were washed three times with pre-cooled PBS and fixed by 3.7% paraformaldehyde for 10 min at room temperature, followed by permeablization for 10 min at room temperature with 0.5% Triton X-100. After incubation for 1 h with 3% BSA at 37 °C, primary antibodies were added for incubation for 1 h at 37 °C. The bound antibodies were visualized by incubation with secondary antibodies (Alexa Fluor 546 donkey anti-mouse IgG). Images were acquired using Zeiss LSM800.

## Chromatin immunoprecipitation (ChIP)
ChIP experiments were carried out on HEK 293 T cells and HepG2-NTCP cells. After treatment as indicated, cells were fixed with 1% formaldehyde (Sigma Aldrich 47608) for 10 min at 25 °C before quenching with 125 mM glycine. Cells were rinsed twice with ice cold PBS and treated by trypsin for 5–10 min, then collected by centrifugation at 400 g for 5 min. Cells were resuspended and lysed for 10 min at 4 °C in Nuclei Lysis Buffer supplemented with protease inhibitors. After chromatin sonication and centrifugation at 13,800 g for 20 min at 4 °C, the supernatant was aliquoted and subjected to immunoprecipitation overnight at 4 °C using 4 μg of H3K4me3, H3K9me3, flag and IgG antibodies, respectively. Immune complexes were incubated for 6 h with 50 μl of a mix of Protein G agarose at 4 °C. Then the beads were sedimented by centrifugation at 960 g for 5 min at 4 °C and washed with Low Salt IP Wash Buffe, High Salt IP Wash Buffer, LiCl IP Wash Buffer and twice with TE buffer. Bound protein-DNA complexes were released from the resins by incubation for 20 min at 65 °C in 200 μl IP Elution buffer for twice and added with 1 v of 1 M NaHCO$_3$ solution, 1/20 v of 5 M NaCl solution (Sigma-Aldrich). DNA crosslinks were reversed by overnight incubation at 65 °C followed by addition of 8 μl 10 mg/ml RNase A at 37 °C for 1 h and 8 μl 10 mg/ml proteinase K at 55 °C for 2 h. Samples were extracted, ethanol precipitated and resuspended in TE buffer. The input DNA treated identically and the recovered DNA were quantified by real-time PCR runs using the Bio-Rad CFX96 Real-time PCR System. The values were calculated as the ratios between the ChIP signals and the respective input DNA signals. The primer sequences used for ChIP-qPCR analyses are listed in the Supplementary Table 3.

## Antibodies
The following antibodies were used in this study: anti-Spindlin1 Rabbit mAb (Cell Signaling Technology, Cat. #: 89139, clone name: E6R1Z, Dilution: 1:1000 for WB and 1:50 for ChIP); anti-DDB1 Rabbit mAb (Cell Signaling Technology, Cat. #: 6998, clone name: D4C8, Dilution: 1:1000 for WB); anti-HA Rabbit mAb (Cell Signaling Technology, Cat. #: 3724, clone name: C29F4, Dilution: 1:2000 for WB and 1:50 for Co-IP); anti-H3K9me3 Rabbit pAb (Abcam, Cat. #: ab8898, Dilution: 1:50 for ChIP); anti-Flag Mouse mAb (Sigma-Aldrich, Cat. #: F1804, clone name: M2, Dilution: 1:2000 for WB and 1:100 for ChIP); anti-β-tubulin Mouse mAb (EarthOx, Cat. #: E021040, Dilution: 1:4000 for WB); anti-H3K4me3 Rabbit mAb (Merck Millipore, Cat. #: 04-745, clone name: MC315, Dilution: 1:50 for ChIP); anti-SMC6 Mouse mAb (abcepta, Cat. #: AT3956a, clone name: 2E7, Dilution: 1:1000 for WB); Alexa Fluor 546 donkey anti-

mouse IgG (Invitrogen, Cat. #: A10036, Dilution: 1:1000 for IF); anti-HBcAg antibody (1C10) was developed and validated by Prof. Wenhui Li's lab; anti-Flag M2 Agarose beads (Sigma-Aldrich, Cat. #: A2220).

## Reporting summary
Further information on research design is available in the Nature Portfolio Reporting Summary linked to this article.

## Data availability
The atomic coordinates and structure factors for the reported complex structure of Spindlin1-HBx have been deposited in the Protein Data bank with accession number 8GTX [https://www.rcsb.org/structure /unreleased/8GTX]. The PDB codes of the previously determined structures used in this manuscript are: 4MZG (Spindlin1-H3K4me3), 7E9M (Spindlin1-SPINDOC$_{256-281}$), 7BQZ (Spindlin1-H3"K4me3-K9me3"), 3I7H (DDB1-HBx$_{88-100}$), 5FCG (Bcl-2-HBx$_{110-135}$), 2NS2 (Spindlin1), 4MZF (Spindlin1-H3 "K4me3-R8me2a"), 7BU9 (Spindlin1-H3 "K4me3-K9me2"), 5Y5W (Spindlin1-H4K20me3). The authors declare that all the data supporting the findings of this study are either shown in the main and supplementary text. Source data are provided with this paper.

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

## Acknowledgements

We thank members of the Haitao Li laboratory for scientific input throughout the study. We thank the staff members at beamline BL18U of the Shanghai Synchrotron Radiation Facility and Dr S.F. at the Tsinghua Center for Structural Biology for their data collection assistance and the China National Center for Protein Sciences in Beijing for facility support. This research was supported by grants from the National Natural Science Foundation of China (92153302 to H.L., 32000447 to W.L., 81525018 to W.H.L., 81773030 to D.W.). National Key R&D Program of China (2021YFA1300103 and 2020YFA0803303 to H.L.). Science and Technology Major Project of Beijing grant (D171100003117003 to W.H.L.). W.L. is a post-doctoral fellow supported by Tsinghua-Peking Center for Life Sciences.

## Author contributions

H.L. conceived the study; W.L. performed biochemical and structural studies with helps from X.S., Y.D., M.Y. F.Z., C.D., X.Z and J.Z.; W.L. and Q.Y. performed cellular studies with help from B.P.; H.L. and W.L. analyzed the data and wrote the manuscript with critical inputs from W.H.L. and D.W.

## Competing interests

The authors declare no competing interests.
