## [Peer Review File · Nature Communications]

nature portfolio

Peer Review FileReviewer comments, first round

Reviewer #1 (Remarks to the Author):

Review of HBV Liu et al

Liu and colleagues report the identification of a novel HBV effector protein interaction motif that binds to the transcription regulator spindlin 1 and modulates HBV transcription. In a series of elegant experiments the authors first map the interaction site on HBx with spindlin 1, then determine the crystal structure of the complex, and dissect the structural basis of the interaction. Then they show the functional consequences of this interaction, and mechanism of action. Overall I very much enjoyed the manuscript, and the line of enquiry pursued by the time. I only have a few minor comments to make:

The different binding modes between spindoc and HBx are intriguing, and should be discussed in a little more detail. Are there significant differences in the precise amino acids that contact the spindlin binding groove between the two interacting peptides? Could the authors speculate as to why the modes are so different?

For all SPR and ITC interactions shown, please state in the appropriate figure legend how often each measurement was performed for error quantification.

Minor issues:

Line 85: "a titling array of 20-mer" – overlapping array?

Line 98: comparison

Line 103: N-terminus of HBx

Line 107: determine the molecular basis or gain molecular insight into

Line 112: The authors state that "HBx2-21 is intimately associated with a hydrophobic groove of Spindlin1 with high degree of shape complementarity". Please report a value for shape complementarity to support this statement, or clarify how shape complementarity was measured.

Figure 2e: Please colour the spindoc chain in a different colour, it is very difficult to compare the binding modes of HBx and spindoc.

Line 122: is also warranted – is supported?

Line 277: pGEX-6p – which one? P1,2 or 3

Line 313: on a large scale - ?

Line 574: Data represent the mean±SD – how many measurements were performed (n=?)

Reviewer #2 (Remarks to the Author):

Weo Liu and colleagues investigate the mechanism by which the HBV regulatory protein HBx activates transcription and replication of the viral nuclear covalently closed circular (ccc) DNA

minichromosome. They show that HBx accomplishes its task by hijacking the epigenetic reader Spindlin1. Through biochemical and structural studies, they show that the N-terminal 2-21 segment of HBx (HBx 2-21) binds Tudor 3 domain of Spindlin1 that this interaction is accompanied by the switch from the repressive H3K9me3 mark to the H3K4me3 mark of active chromatin and promotes transcription from the cccDNA.

The work is well conceived, and the AA present a set of high-quality results. However, the AA fail to clarify several important points:

i) how the interaction between HBx and Spindlin-1 relates with the well-established activity of HBx (binding to DDB1 to redirect the DDB1 containing E3 ubiquitin ligase complex to degrade Smc 5/6). Is HBx-Spindlin1 interaction downstream or upstream HBx activity on Smc 5/6? or, in other words, is HBx-Spindlin1 interaction required for HBx binding/activity on DDB1, or vice versa?

ii) these considerations lead to another important question. From Figure 5, it appears that the effect on HBV transcription/replication of abrogating Spindlin1 activity is, to say the least, partial. In Figure 5G, the AA show that the Spindlin1 binding to the HBV cccDNA is well preserved in cells infected by the HBV HBx- mutant. Thus, the contribution of Spindlin1 to HBV transcription is there but modest and HBx is not required for Spindlin1 binding to the cccDNA. These considerations do not question the main message of the paper that HBx interaction with Spindlin1 has some effect on HBV cccDNA transcription. However, the AA must put their observations in the context of all other mechanisms that have been reported, repeatedly, to affect the epigenetic status of the HBV minichromosome and cccDNA transcription and to do so by generating novel experimental data. iii) the AA do not provide any direct mechanistic insight on how the interaction between HBx and Spindlin1 promotes the switch from the repressive H3K9me3 mark to the H3K4me3 mark of active chromatin on the cccDNA-bound histones.

Finally, all functional data are obtained in HBV-infected HepG2-NTCP cells. The AA must confirm the observations (and their robustness) in HBV-infected primary human hepatocytes.

As a minor point, the AA state "HBx interacts with Bcl-xL through the BH3-like domain to promote HBV production". It is unclear how this interaction relates to HBx activity on cccDNA transcription.

Reviewer #3 (Remarks to the Author):

Spindlin1, an epigenetic reader and transcriptional coactivator, has previously been shown to interact with the HBV X protein (HBx), thereby modulating HBV gene transcription. However, the underlying mechanism was not clearly defined. In this manuscript, Liu et al. provide evidence that the host protein Spindlin1, can directly interact with an N-terminal motif within HBx. This interaction appears to facilitate binding to HBV cccDNA to promote HBV transcription in an H3K4me3/H3K9me3-dependent manner. Using biochemical assays the authors explored systematically the interplay between Spindlin1 and HBx and confirmed their findings using x-ray crystallography.

Overall, this is a well-designed study which provides new insights into how HBV hijack host factors, here Spindlin-1 to sustain establish and maintain persistence through epigenetic regulation. The data are clearly presented and largely support the authors' conclusions. However, there are a few concerns and open questions regarding the interplay between Spindlin1-HBx and how this complex interacts with cccDNA that need to be further addressed.

Major comments

- The authors have mapped amino acids that appear to be important for Spindlin1 and HBx binding (Figures 2&3) *in vitro*. It will be helpful to test the binding affinity between Spindlin1V232R, Spindlin1I245R, Spindlin1K216S/V218R and HBx inside of the cells, particularly in the hepatic cells (for instance HepG2 or Huh7).
- For the Spindlin1-HBx interaction with cccDNA part: The authors need to validate that the HBV DNA enriched with an anti-Spindlin1 is primarily cccDNA, and not HBV rcDNA, dsDNA or even integrated HBV DNA? Did the authors perform a T5 digestion step after CHIP and before qPCR?
- Which region of the HBV genome does the Spindlin1-HBx complex bind to? Is it BCP region or other HBV transcription regulation regions like En 1/2?
- How conserved the region within HBx (2-21) which appears to facilitate binding to Spindlin1 across different HBV genotypes?
- In figure 1f, the bands for HBx2-21M are larger than that for WT. Did the authors compare the

molecular weight of these two proteins to confirm this?

- For the ChIP experiments in figure 4e, it would be important to also show the DNA gel and Sanger sequencing data to confirm the protein-DNA binding.
- Line 144, style avoid using "interestingly", rather use notably or similar
- line 159, "hepatocytes" rather than "hepatocyte"
- Spindlin1 is thought to be a proto-oncogene which can promote oncogenic transcription and cell proliferation. Have the authors tested Spindlin1 knockdown has any effects on the cell cycle? Would the authors expect any effects on non-dividing (primary) hepatocytes

Response Letter for “Molecular insights into Spindlin1-HBx interplay and its impact on HBV transcription from cccDNA minichromosome” (MS ID#: NCOMMS-22-48262A)

Reviewer #1

Remarks to the Author:

Liu and colleagues report the identification of a novel HBV effector protein interaction motif that binds to the transcription regulator spindling 1 and modulates HBV transcription. In a series of elegant experiments the authors first map the interaction site on HBx with spindlin 1, then determine the crystal structure of the complex, and dissect the structural basis of the interaction. Then they show the functional consequences of this interaction, and mechanism of action. Overall I very much enjoyed the manuscript, and the line of enquiry pursued by the time. I only have a few minor comments to make:

Authors’ response: We thank this reviewer for his/her positive comments by stating that “Overall I very much enjoyed the manuscript, and the line of enquiry pursued by the time.”

The different binding modes between spindoc and HBx are intriguing, and should be discussed in a little more detail. Are there significant differences in the precise amino acids that contact the spindling binding groove between the two interacting peptides? Could the authors speculate as to why the modes are so different?

Authors’ response: Thanks for the comments. Structural alignment has revealed that HBx₂₋₂₁ interacts with Tudor 3 of Spindlin1 in an opposite N to C orientation compared to that of SPINDOC₂₅₆₋₂₈₁ (Fig. 2e). Despite this, both HBx₂₋₂₁ and SPINDOC₂₅₆₋₂₈₁ complete the β -barrel fold of Tudor 3 with similar hydrophobic core formation and β -sheet formation (Figure R1a). This highlights both the conservation and diversity of Spindlin1’s Tudor 3-mediated partner engagement.

Upon complex formation, HBx₂₋₂₁ triggered minimal conformational change of Tudor 3, while SPINDOC₂₅₆₋₂₈₁ induced more pronounced structural rearrangement around β strands 1 and 2 of Tudor 3 (Figure R1b, c). One reason for the observed difference is the unique extended N-terminal motif of SPINDOC₂₅₆₋₂₈₁, starting from F256, which contributes to binding (Figure R1c). We have included this detail discussions in our revised manuscript as new Supplementary Fig. 2.

Figure R1. Structural comparison of Spindlin1-HBx₂₋₂₁ complex with Spindlin1-SPINDOC₂₅₆₋₂₈₁ complex. (a) Both HBx₂₋₂₁ (light magenta) and SPINDOC₂₅₆₋₂₈₁ (cyan) complete the β -barrel fold of Tudor 3 with similar hydrophobic core formation and β -sheet formation. (b) Structural alignment of free Spindlin1 Tudor 3 (green) (PDB ID 4MZF) and Spindlin1-HBx₂₋₂₁ complex (b, pink-yellow) or Spindlin1-SPINDOC₂₅₆₋₂₈₁ (c, violet-slate, PDB ID 5B1Z).

For all SPR and ITC interactions shown, please state in the appropriate figure legend how often each measurement was performed for error quantification.

Authors' response: Thanks for the comments. An ITC curve typically involves more than 10 successive titrations (17 in this paper). The binding K_d and other thermodynamic parameters are calculated by fitting the titration curve using nonlinear regression. In this process, each data point represents an independent measurement of protein/ligand concentrations and a set of thermodynamic parameters, and the standard error of each parameter (e.g. K_d) will be estimated based on curve fitting over all data points (n=17 here) at 95% confidence. Similarly, for SPR analysis, signals were collected at one point/second (average of 8 images), and a total of 1,000 points were collected. The standard errors of binding parameters are calculated by fitting all data points as stated in the method section.

Minor issues:

Line 85: "a titling array of 20-mer" – overlapping array?

Line 98: comparison

Line 103: N-terminus of HBx

Line 107: determine the molecular basis or gain molecular insight into

Authors' response: Corrected. Thanks a lot!

Line 112: The authors state that "HBx₂₋₂₁ is intimately associated with a hydrophobic groove of Spindlin1 with high degree of shape complementarity". Please report a value for shape complementarity to support this statement, or clarify how shape complementarity was measured.

Authors' response: Thanks for the suggestion. The revised manuscript has included Sc (surface complementarity) score to quantitatively describe the shape complementarity between HBx and Spindlin1. We calculated a Sc score of 0.7, which is even higher than that of antibody-antigen interfaces (where Sc ranges from 0.64 to 0.68) (Lawrence and Colman, 1993)"

Lawrence, M. C. & Colman, P. M. Shape Complementarity at Protein-Protein Interfaces. *Journal of Molecular Biology* **234**, 946-950, doi:DOI 10.1006/jmbi.1993.1648 (1993).

Figure 2e: Please colour the spindoc chain in a different colour, it is very difficult to compare the binding modes of HBx and spindoc.

Authors' response: Thanks for the valuable suggestion. We have recolored the SPINDOC chain accordingly (Figure R2). The updated figure (Fig. 2e) has been included in the revised

manuscript.

Figure R2. Cartoon representation of structural comparison of Spindlin1-HBx₂₋₂₁ complex with Spindlin1-SPINDOC₂₅₆₋₂₈₁ complex (PDB ID 5B1Z, light grey-slate).

Line 122: is also warranted – is supported?

Authors' response: Corrected! Thanks.

Line 277: pGEX-6p – which one? P1,2 or 3

Authors' response: We apologize for this confusion. We have labeled the plasmid name (pGEX-6P-1) in our revised manuscript.

Line 313: on a large scale - ?

Authors' response: Thanks, we have rephrased this sentence to “Peptide concentrations were determined by weighing in large quantities”.

Line 574: Data represent the mean±SD – how many measurements were performed (n=?)

Authors' response: Thanks for the comment. The data was presented as mean±SE (standard error), which is estimated by nonlinear regression of the titration curve at 95% confidence. The data points used for fitting is 16 (n=16).

Reviewer #2

Remarks to the Author:

Wei Liu and colleagues investigate the mechanism by which the HBV regulatory protein HBx activates transcription and replication of the viral nuclear covalently closed circular (ccc) DNA minichromosome. They show that HBx accomplishes its task by hijacking the epigenetic reader Spindlin1. Through biochemical and structural studies, they show that the N-terminal 2-21 segment of HBx (HBx 2-21) binds Tudor 3 domain of Spindlin1 that this interaction is accompanied by the switch from the repressive H3K9me3 mark to the H3K4me3 mark of active chromatin and promotes transcription from the cccDNA.

The work is well conceived, and the AA present a set of high-quality results.

Authors' response: We thank this reviewer for his/her positive comments by stating that "*The work is well conceived, and the AA present a set of high-quality results.*"

However, the AA fail to clarify several important points:

i) how the interaction between HBx and Spindlin-1 relates with the well-established activity of HBx (binding to DDB1 to redirect the DDB1 containing E3 ubiquitin ligase complex to degrade Smc 5/6). Is HBx-Spindlin1 interaction down stream or upstream HBx activity on Smc 5/6? or, in other words, is HBx-Spindlin1 interaction required for HBx binding/activity on DDB1, or vice versa?

Authors' response: Thanks for raising this important question. HBx binds to DDB1 through an H-box motif, HBx₈₈₋₁₀₀, which is distinct from HBx₂₋₂₁. Therefore, Spindlin1 and DDB1 do not directly compete for HBx binding. However, the predicted structure of HBx by AlphaFold (Jumper et al., 2021) shows that elements HBx₂₋₂₁, HBx₈₈₋₁₀₀ and HBx₁₀₄₋₁₃₅ cluster to form a hydrophobic core (Figure R3a). Interestingly, in the folded HBx structure, key residues involved in Spindlin1, DDB1 and Bcl-2 binding, such as M5, L16, L18 of HBx₂₋₂₁, L89, L93, R96 of HBx₈₈₋₁₀₀ and V116, W120, L123, I127, R128 of HBx₁₁₀₋₁₃₅, are buried upon hydrophobic core formation (Figure R3a). This suggests that effective engagement of HBx with DDB1 and Spindlin1 requires unfolding of HBx to expose key binding motifs. In this case, we speculate that HBx may exist in two conformational states: a folded inactive state and an extended active state (Figure R3b). Conceivably, Spindlin1, DDB1 and Bcl-2 may cooperate with each other to interact with HBx by jointly overcoming the folding energy barrier, thus enabling a functional switch of HBx from an inactive state to an active one.

To experimentally investigate the functional correlation between Spindlin1 and DDB1, we conducted HBx pulldown and SMC6 degradation assays in HepG2-NTCP cells. We showed that siRNA knockdown of Spindlin1 resulted in slightly decreased level of DDB1 that binds to HBx (Figure R3c). In SMC6 degradation assays, our western blot analysis revealed a significant decrease in the SMC6 level in cells expressing HBx. Surprisingly, such a decrease was partly restored in Spindlin1 knockdown cells (Figure R3d). These observations are consistent with our structural analysis above, and support a positive crosstalk between Spindlin1-HBx and DDB1-HBx engagements as well as DDB1-HBx-mediated SMC5/6 degradation. We have included

these new results in the revised manuscript as new Fig.7.

Figure R3. Crosstalk between Spindlin1-HBx and DDB1-HBx engagements. (a) AlphaFold model of free HBx (left) and structure model of HBx binding to Spindlin1 (light magenta-green), DDB1 (cyan-grey, PDB ID 3I7H) and Bcl-2 (slate-white, PDB ID 5FCG). (b) Schematic model of HBx transitioning from an inactive state to an active one by binding to Spindlin1, DDB1, and Bcl-2. (c) Co-IP of HA-HBx with endogenous DDB1 using anti-HA antibodies in HepG2-NTCP cells, which were transfected with Ctrl siRNA or Spindlin1 siRNA 1. (d) WB analyzing of the SMC6 level in Spindlin1 knockdown HepG2-NTCP cells expressing HBx. Cells were transfected with Ctrl siRNA or Spindlin1 siRNA 1 for 72h and then transfected with HA-HBx or empty vector. After 96h, cells were harvest for WB analysis.

Jumper, J. *et al.* Highly accurate protein structure prediction with AlphaFold. *Nature* **596**, 583-589, doi:10.1038/s41586-021-03819-2 (2021).

ii) these considerations lead to another important question. From Figure 5, it appears that the effect on HBV transcription/replication of abrogating Spindlin1 activity is, to say the least, partial. In Figure 5G, the AA show that the Spindlin1 binding to the HBV cccDNA is well preserved in cells infected by the HBV HBx- mutant. Thus, the contribution of Spindlin1 to HBV transcription is there but modest and HBx is not required for Spindlin1 binding to the cccDNA. These considerations do not question the main message of the paper that HBx interaction with Spindlin1 has some effect on HBV cccDNA transcription. However, the AA must put their observations in the context of all other mechanisms that have been reported, repeatedly, to affect the epigenetic status of the HBV minichromosome and cccDNA transcription and to do so by generating novel experimental data.

Authors' response: Thanks for the valuable suggestion. Our study on HBV infected HepG2-

NTCP and primary human hepatocytes demonstrated that inhibiting Spindlin1 activity resulted in a significant reduction in HBV transcription and HBeAg levels (Fig.5b-e, Supplementary Fig.4a-f). Rescue experiments also revealed that the ectopic expression of wild type Spindlin1 can restore HBV transcription and HBeAg levels in Spindlin1 knockdown cells (Fig. 5f, g), indicating that Spindlin1 plays a crucial role in promoting HBV transcription. While our findings are reliable and reproducible, we acknowledge that the role of Spindlin1-HBx engagement on HBV transcription/replication represents only one of the many mechanisms involved in HBV regulation. We are pleased that our work has identified a previously uncharacterized mechanism, highlighting the regulatory complexity of HBV life cycle from the perspective of epigenetics and cccDNA minichromosome.

In Fig. 5g, our ChIP analysis revealed a clear signal of Spindlin1 on cccDNA in HepG2-NTCP cells infected with wild type HBV. In contrast, the enrichment of Spindlin1 was reduced when HBx-deficient HBV was used for infection, indicating that HBx plays a role in stabilizing Spindlin1 at the HBV minichromosome. However, the enrichment of Spindlin1 on cccDNA still exists compared to the anti-IgG group (~0.5% of input), suggesting that Spindlin1 could target the cccDNA minichromosome via an HBx-independent mechanism. In fact, this is consistent with our observation that Spindlin1 is a potent reader of the histone H3 “K4me3-K9me3” methylation pattern ($K_d = 13.3$ nM) that marks poised heterochromatic regions (Du et al., 2021; Zhao et al., 2020a). Our binding studies revealed that when Spindlin1 forms a complex with HBx, its binding to H3 “K4me3-K9me3” could be further enhanced from 13.3 nM to 3.8 nM (Fig. 4b). Importantly, as discussed early, in addition to recruitment, Spindlin1-HBx engagement also plays a critical role in switching the conformational state of HBx, thereby unlocking its function. In the revised manuscript, we have included a new Fig. 7 to explain these findings and provide more discussion to contextualize our new observations with other reported mechanisms.

Zhao, F. *et al.* Molecular basis for histone H3 "K4me3-K9me3/2" methylation pattern readout by Spindlin1. *J Biol Chem* **295**, 16877-16887, doi:10.1074/jbc.RA120.013649 (2020).

Du, Y. *et al.* Structural mechanism of bivalent histone H3K4me3K9me3 recognition by the Spindlin1/C11orf84 complex in rRNA transcription activation. *Nat Commun* **12**, 949, doi:10.1038/s41467-021-21236-x (2021).

iii) the AA do not provide any direct mechanistic insight on how the interaction between HBx and Spindlin1 promotes the switch from the repressive H3K9me3 mark to the H3K4me3 mark of active chromatin on the cccDNA-bound histones.

Authors' response: Thank you for the suggestion. As previously discussed, we have proposed in the revised manuscript a mechanism for the conformational switch of HBx triggered by Spindlin1 and DDB1. This explains how Spindlin1-HBx interaction initiates a switch of the cccDNA chromatin state from a repressive to an active one.

In mammalian cells, the SET1/MLL complexes are the primary writers of the H3K4me3 mark. It has been reported that they can methylate H3K4 in the presence of H3K9me3 (Patel et al., 2014). Therefore, H3K4me3 can be enzymatically created in H3K9me3-rich regions to establish a bivalent H3 "K4me3-K9me3" methylation pattern for robust Spindlin1 recruitment.

In this process, engagement between Spindlin1 and HBx not only promotes Spindlin1 recruitment but also triggers a functional switch of HBx from organizing a repressive chromatin state to an open one by cooperatively recruiting activation-related factors such as DDB1, CBP/p300, or other HBV-specific transcription factors (TFs). Intriguingly, it has been well established that erasers of H3K9me2/3, such as KDM4 and KDM7 family members, can recognize H3K4me3 to remove H3K9me2/3 marks. This "read-erase" mechanism may function to resolve the H3 "K4me3-K9me3" bivalent state and establish a fully active chromatin state marked by H3K4me3 and hyperacetylation. We have added more discussions and an updated working model (new Fig. 7e) to illustrate these points in the revised manuscript. Further research will be conducted to experimentally investigate whether and how the above-mentioned factors, such as SET1/MLL, KDM4/7, CBP/p300, participate in the epigenetic reprogramming of cccDNA minichromosome initiated by Spindlin1-HBx engagement.

Patel, A. *et al.* Automethylation activities within the mixed lineage leukemia-1 (MLL1) core complex reveal evidence supporting a "two-active site" model for multiple histone H3 lysine 4 methylation. *J Biol Chem* **289**, 868-884, doi:10.1074/jbc.M113.501064 (2014).

Finally, all functional data are obtained in HBV-infected HepG2-NTCP cells. The AA must confirm the observations (and their robustness) in HBV-infected primary human hepatocytes.

Authors' response: Thanks for the suggestion. While HepG2-NTCP cells are an excellent model for studying HBV infection, we acknowledge the importance of confirming the observations and their robustness in HBV-infected primary human hepatocytes. Therefore, we conducted Spindlin1 knockdown studies in HBV-infected primary human hepatocytes. As anticipated, knockdown of Spindlin1 (Figure R4a) led to a significant reduction in HBV RNA levels (Figure R4b) and HBeAg levels in the medium supernatant (Figure R4c). These results are consistent with those obtained in HBV-infected HepG2-NTCP cells, indicating the crucial role of Spindlin1 in stimulating HBV transcription. These new results have been incorporated in the revised Supplementary Fig. 4d-f.

Figure R4. Spindlin1 is required for HBV transcription in HBV-infected primary human hepatocytes. (a) The efficiency of Spindlin1 knockdown in HBV infected cells by RT-qPCR

analysis. (b-c) Primary human hepatocytes were transfected with Ctrl siRNA or Spindlin1 siRNAs for 16h and then infected with HBV. At 11dpi, cells and culture medium were harvest and HBV RNA levels and HBeAg levels were analyzed by RT-qPCR (b) and ELISA (c), respectively.

As a minor point, the AA state "HBx interacts with Bcl-xL through the BH3-like domain to promote HBV production". It is unclear how this interaction relates to HBx activity on cccDNA transcription.

Authors' response: Thanks for the comment. The BH3-like domain spans residue 110-135 of HBx, which overlaps with the HBX₁₀₄₋₁₃₅ fragment that forms the hydrophobic core together with HBX₂₋₂₁ and HBX₈₈₋₁₀₀. Conceivably, Bcl2 or Bcl-xL, DDB1 and Spindlin1 may cooperatively bind to an unfolded state HBx to bring about an open chromatin state of cccDNA for active transcription or replication. We have included this discussion in the revised manuscript.

Reviewer #3

Remarks to the Author:

Spindlin1, an epigenetic reader and transcriptional coactivator, has previously be shown to interact with the HBV X protein (HBx), thereby modulating HBV gene transcription. However, the underlying mechanism was not clearly defined. In this manuscript, Liu et al. provide evidence that the host protein Spindlin1, can directly interact with an N-terminal motif within HBx. This interaction appears to facilitate binding to HBV cccDNA to promote HBV transcription in an H3K4me3K9me3- dependent manner. Using biochemical assays the authors explored systematically the interplay between Spindlin1 and HBx and confirmed their findings using x-ray crystallography.

Overall, this is a well-designed study which provides new insights into how HBV hijack host factors, here spindling-1 to sustain establish and maintain persistence through epigenetic regulation. The data are clearly presented and largely support the authors' conclusions. However, there are a few concerns and open questions regarding the interplay between Spindlin1-HBx and how this complex interacts with cccDNA that need be further addressed.

Authors' response: We thank this reviewer for his/her recognition of our work by stating that “Overall, this is a well-designed study which provides new insights into...”and “The data are clearly presented and largely support the authors' conclusions.”

Major comments

- *The authors have mapped amino acids that appear to be important for Spindlin1 and HBx binding (Figures 2&3) in vitro. It will be helpful to test the binding affinity between Spindlin1V232R, Spindlin1I245R, Spindlin1K216S/V218R and HBx inside of the cells, particularly in the hepatic cells (for instance HepG2 or Huh7).*

Authors' response: Thanks for the comments. Based on the suggestion of this reviewer, we conducted Co-IP assays in Huh7 cells to evaluate the interaction between HBx and Spindlin1 mutants. Full-length Spindlin1 with single point mutations, V232R and I245R, did not express well in a cellular context, while the double mutant, K216S/V218R, worked fine. As shown in Figure R5a, the binding of Spindlin1 K216S/V218R with HBx was significantly compromised compared to wild type Spindlin1, which is consistent with our *in vitro* ITC titration and cell cultural rescue experiments. The new result has been incorporated in the revised Supplementary Fig. 4g. The residue binding likely reflects other indirect binding effects. For instance, our unpublished data demonstrate that the N-terminal flexible tail (1-49) of Spindlin1 could mediate its self-association and liquid-liquid phase separation (Figure R5b).

Figure R5. Immunoprecipitation assay in Huh7 cells to assess the interaction between HBx and Spindlin1 mutant. (a) Cells were co-transfected with HA-tagged HBx and wild type Flag-Spindlin1 or K216S/V218R double mutant Spindlin1. After 48h, cellular extracts were immunoprecipitated with anti-Flag and anti-HA antibodies and analyzed by the indicated antibodies using WB. β -tubulin was measured and analyzed as an input control. (b) Domain architecture of Spindlin1 (upper). N-terminal flexible tail (1-49) of Spindlin1 mediates its self-association and liquid-liquid phase separation in vivo and in vitro (down). Scale bars: 10 μ m. (Unpublished data).

- For the Spindlin1-HBx interaction with cccDNA part: The authors need to validate that the HBV DNA enriched with an anti-Spindlin1 is primarily cccDNA, and not HBV rcDNA, dsIDNA or even integrated HBV DNA? Did the authors perform a T5 digestion step after CHIP and before qPCR?

Authors' response: We appreciate this reviewer's valuable comments. As indicated Fig. 5a, our ChIP experiments on HepG2-NTCP cells infected with HBV revealed that Spindlin1 was concentrated at the HBV cccDNA minichromosom. While doing the ChIP experiment, the cccDNA specific primers (HBV cccDNA-F: 5'-GTGCACTTCGCTTCACCTCT-3', positions 1579-1598; HBV cccDNA-R: 5'-AGCTTGGAGGCTTGAACAGT3', positions 1859-1878; positions were indicated relative to the EcoRI site.) were used to exclude the HBV rcDNA, dsIDNA and integrated HBV DNA as previous reported (Benhenda et al., 2013; Decorsiere et al., 2016; Riviere et al., 2015). Previous studies have shown that, in HBV-infected HepG2-NTCP cells, the cccDNA selective primers targeting this region can effectively exclude the rcDNA, dsIDNA and integrated HBV DNA in nucleus, and the DNA signal is primarily from cccDNA (Tropberger, P. et al., 2015).

We agree with the reviewer that T5 digestion after ChIP and before qPCR may further remove contaminants of other forms of HBV DNA. However, the total amount of purified DNA for ChIP-qPCR is low, further processing with T5 including the purification step likely introduces more random variables than the contaminating HBV DNA itself. Therefore, we followed the conventional steps in the field using the selective cccDNA primers and with no T5 digestion in the process.

Benhenda, S. *et al.* Methyltransferase PRMT1 is a binding partner of HBx and a negative regulator of hepatitis B virus transcription. *J Virol* **87**, 4360-4371, doi:10.1128/JVI.02574-12 (2013).

Riviere, L. *et al.* HBx relieves chromatin-mediated transcriptional repression of hepatitis B viral cccDNA involving SETDB1 histone methyltransferase. *J Hepatol* **63**, 1093-1102, doi:10.1016/j.jhep.2015.06.023 (2015).

Decorsiere, A. *et al.* Hepatitis B virus X protein identifies the Smc5/6 complex as a host restriction factor. *Nature* **531**, 386-389, doi:10.1038/nature17170 (2016).

Tropberger, P. *et al.* Mapping of histone modifications in episomal HBV cccDNA uncovers an unusual chromatin organization amenable to epigenetic manipulation. *Proceedings of the National Academy of Sciences* **112**, E5715-E5724 (2015).

• Which region of the HBV genome does the Spindlin1-HBx complex bind to? Is it BCP region or other HBV transcription regulation regions like En 1/2?

Authors' response: Thanks for the comments. In our ChIP-qPCR assays, we used primers (HBV cccDNA-F: 5'-GTGCACTTCGCTTCACCTCT-3', positions 1579-1598; HBV cccDNA-R: 5'-AGCTTGGAGGCTTG AACAGT3', positions 1859-1878) that cover the BCP region (positions 1742-1849) and En2 region (positions 1685-1773) of the HBV DNA (Figure R6a). These primers have been regarded as specific to cccDNA and are frequently used in ChIP-qPCR studies of HBV episomes (Benhenda *et al.*, 2013; Riviere *et al.*, 2015; Decorsiere *et al.*, 2016). We observed a significant increase in signal in the anti-Spindlin1 (Figure R6b), anti-H3K4me3 (Figure R6c), and anti-H3K9me3 ChIP assays (Figure R6d), indicating that the Spindlin1-HBx complex binds to a chromatin context across the BCP and En2 regions of HBV cccDNA.

Figure R6. (a) Structure of the HBV genome with the open reading frames (ORFs) shown as curved arrows, the CBP region (yellow) and enhancer2 region (red). (a-d) ChIP experiments using HBV infected cells showed Spindlin1 (a), H3K4me3 (b) and H3K9me3 (c) is recruited to the HBV cccDNA BCP and En2 region. HepG2-NTCP cells were infected with HBV and were harvested and analyzed by ChIP using antibodies against Spindlin1, H3K4me3 and H3K9me3 at 7dpi.

Benhenda, S. *et al.* Methyltransferase PRMT1 is a binding partner of HBx and a negative regulator of hepatitis B virus transcription. *J Virol* **87**, 4360-4371, doi:10.1128/JVI.02574-12 (2013).

Riviere, L. *et al.* HBx relieves chromatin-mediated transcriptional repression of hepatitis B viral cccDNA

involving SETDB1 histone methyltransferase. *J Hepatol* **63**, 1093–1102, doi:10.1016/j.jhep.2015.06.023 (2015).

Decorsiere, A. *et al.* Hepatitis B virus X protein identifies the Smc5/6 complex as a host restriction factor. *Nature* **531**, 386–389, doi:10.1038/nature17170 (2016).

• How conserved the region within HBx (2-21) which appears to facilitate binding to Spindlin1 across different HBV genotypes?

Authors' response: The region within HBx (2-21) is highly conserved, especially the key residues involved in Spindlin1 engagement (annotated by *). As shown in the sequence alignment in Figure R7a, the HBx (2-21) region from ten human HBV genotypes (A to J), woodchuck (WHV), ground squirrel (GSHBV), bat HBV (BHBV), and woolly monkey HBV (WMHBV) shares 55% sequence identity. We have also conducted ITC assays and confirmed micromolar level binding between Spindlin1 and other representative HBV genotypes (Figure R7b), highlighting the conserved function of Spindlin1 in promoting viral gene expression among a wide range of hepadnaviral family members.

Figure R7. Conservation of Spindlin1-HBx interaction in mammals. (a) Alignment of HBx protein sequences from various human HBV subtypes and mammals. * Key residues involved in Spindlin1 engagement. (b) ITC fitting curves of HBx-D₂₋₂₁, WHx₂₋₂₁ and GSHBV₂₋₂₁ peptides titrated to Spindlin1₅₀₋₂₆₂ protein.

• in figure 1f, the bands for HBX2-21M are larger than that for WT. Did the authors compare

the molecular weight of these two proteins to confirm this?

Authors' response: Thank you for carefully reading our manuscript. The theoretical molecular weights of wild type and mutant HA-HBx proteins are as follows: 18.45 KDa for WT, 17.65 KDa for HA-HBx_{2-21M} (M1), 17.75 KDa for HA-HBx_{11-30M} (M2), and 17.73 KDa for HA-HBx_{61-80M} (M3). In order to confirm their migration positions in SDS-PAGE, we purified the four proteins and observed that the band positions of the four proteins are slightly different, with M2 being the highest, followed by WT, M1, and M3, despite their similar molecular weight (Figure R8a). This is consistent with our WB results (Figure R8b). The slight difference in migration distance likely reflects the variation in amino acid composition among the four proteins. We have included the new SDS-PAGE results in Supplementary Fig. 1d.

Figure R8. (a) SDS-PAGE gel showing the band positions of the four purified proteins: WT, M1, M2 and M3 proteins. (b) WB results showing the bands for WT, M1, M2 and M3.

• For the ChIP experiments in figure 4e, it would be important to also show the DNA gel and Sanger sequencing data to confirm the protein-DNA binding.

Authors' response: Thanks. Based on the reviewer's suggestions, we have added the DNA gel (Figure R9a) and representative Sanger sequencing data (input samples of Spindlin1 group, Figure R9b). Please find the updated figure and caption below, which are included as new Supplementary Fig. 3d, e in the revised manuscript. All the Sanger sequencing data have been provided in the Fig. 4e_Sanger sequencing data folder.

Figure R9. DNA gel (d) and representative Sanger sequencing data (e) of ChIP-qPCR results in HEK 293T cells, which were transfected with Flag-Spindlin1 (S) or Flag-Spindlin1 plus HBx (S+H). The bases marked in red represent the corresponding primer sequences used for qPCR analyses.

• Line 144, style avoid using “interestingly”, rather use notably or similar ()

Authors' response: Corrected!

- *line 159, “hepatocytes” rather than “hepatocyte”*

Authors' response: Corrected!

- *Spindlin1 is thought to be a proto-oncogene which can promote oncogenic transcription and cell proliferation. Have the authors tested Spindlin1 knockdown has any effects on the cell cycle?*

Authors' response: Thanks for the suggestion. As a proto-oncogene, Spindlin1 is overexpressed and promotes oncogenic transcriptional programs in various types of malignant tumors. In 2019, Zhao et al. reported that knockdown of Spindlin1 significantly blocked the cell cycle of liver cancer cells (HepG2 and Huh7) (Zhao et al., 2020). Our laboratory is actively investigating the role of Spindlin1-HBx interaction in promoting tumor progression in liver cancer. We expect to report these results in a separate story.

Zhao, M. *et al.* SPIN1 triggers abnormal lipid metabolism and enhances tumor growth in liver cancer. *Cancer Lett* 470, 54-63, doi:10.1016/j.canlet.2019.11.032 (2020).

- *Would the authors expect any effects on non-dividing (primary) hepatocytes*

Authors' response: Thanks! As explained in our response to Reviewer 2, we depleted Spindlin1 in HBV infected primary human hepatocytes and observed a significant reduction in HBV transcription and HBeAg levels, as shown in Figure R4 (new Supplementary Fig. 4d-f). These results are consistent with our findings in HepG2-NTCP cells infected with HBV.

Reviewer comments, second round

Reviewer #2 (Remarks to the Author):

Liu and Coll have made an excellent job in answering to the reviewers concerns. In particular, they have nicely and thoroughly answered to the 3 main questions I raised: a) how HBx and Spindlin-1 interaction relates or impacts on HBx binding to DDB1 and Smc 5/6 degradation to start/boost cccDNA transcription; b) the role / requirement of HBx for spindlin-1 binding to cccDNA and functions; c) confirm results in HBV-infected PHHs.

Reviewer #3 (Remarks to the Author):

The authors have carefully addressed all the question that I had raised in the previous round of review.

Point-by-point response letter to the reviewers' comments for “Molecular insights into Spindlin1-HBx interplay and its impact on HBV transcription from cccDNA minichromosome” (MS ID#: NCOMMS-22-48262B)

Reviewer #2 (Remarks to the Author):

Liu and Coll have made an excellent job in answering to the reviewers concerns. In particular, they have nicely and thoroughly answered to the 3 main questions I raised: a) how HBx and Spindlin-1 interaction relates or impacts on HBx binding to DDB1 and Smc 5/6 degradation to start/boost cccDNA transcription; b) the role / requirement of HBx for spindlin-1 binding to cccDNA and functions; c) confirm results in HBV-infected PHHs.

Authors' response: We appreciate the reviewer for his/her expert evaluation of our work.

Reviewer #3 (Remarks to the Author):

The authors have carefully addressed all the question that I had raised in the previous round of review.

Authors' response: We appreciate the reviewer for his/her recognition of our work.